# REFINE: A Fine-Grained Medication Recommendation System Using Deep Learning and Personalized Drug Interaction Modeling

**Suman Bhoi**[1]    **Mong Li Lee**[1]    **Wynne Hsu**[1]    **Ngiap Chuan Tan**[2]

[1]National University of Singapore    [2]SingHealth Polyclinics
sumanbhoi@u.nus.edu, {leeml,whsu}@comp.nus.edu.sg, tan.ngiap.chuan@singhealth.com.sg

## Abstract

Patients with co-morbidities often require multiple medications to manage their conditions. However, existing medication recommendation systems only offer class-level medications and regard all interactions among drugs to have the same level of severity. This limits their ability to provide personalized and safe recommendations tailored to individual needs. In this work, we introduce a deep learning-based fine-grained medication recommendation system called REFINE, which is designed to improve treatment outcomes and minimize adverse drug interactions. In order to better characterize patient's health conditions, we model the trend in medication dosage titrations and lab test responses, and adapt the vision transformer to obtain effective patient representations. We also model drug interaction severity levels as weighted graphs to learn safe drug combinations and design a balanced loss function to avoid overly conservative recommendations and miss medications that might be needed for certain conditions. Extensive experiments on two real-world datasets shows that REFINE outperforms state-of-the-art techniques.

## 1   Introduction

The widespread adoption of digital patient records in hospitals has led to the creation of a rich knowledge base in the form of Electronic Health Records (EHR). This valuable resource can be utilized for various tasks in the medical domain, such as mortality prediction [19], outcome prediction [14], and medication recommendation [16, 2]. Concurrently, advances in drug therapy have resulted in growing interest in research aimed at enabling personalized and safe treatment recommendations for managing patient conditions. Current medication recommendation systems are coarse-grained, providing recommendation at the class level rather than individual drugs, and are unable to differentiate between patients who require single versus multiple drugs from the same class. For example, while diabetic patients are often prescribed Metformin to manage their blood sugar levels, patients with uncontrolled blood sugar level may be given additional blood glucose lowering drug such as Glipizide which belong to the same medication class A10B[1] as Metformin. As such, the recommended treatment regime is the same for both types of patients which is suboptimal. Thus, it is essential to develop a fine-grained recommender system capable of providing recommendations at the drug level.

Existing recommender systems that consider drug interactions assume that all the interactions have the same severity. This is not true in practice as resources on drug information such as the DDInter database [23] record different levels of interaction severity, ranging from mild to moderate to severe. By assuming all interactions are equally severe, existing recommender systems tend to avoid any combination of drugs with known interactions, even when the benefits outweigh the potential risks.

---

[1]https://www.whocc.no/atc_ddd_index/?code=A10B&showdescription=no

For example, in the benchmark MIMIC-IV [10], Metformin and Lisinopril are prescribed together 36% of the time for patients with diabetes and hypertension to better manage their conditions, despite having a moderate level of interaction severity.

In this work, we propose a fine-grained medication recommender system that takes into account the severity of drug interactions and leverages information gathered over past visits to provide safe and personalized drug treatment regimes. We represent the varying severity of drug interactions as a weighted graph and employ a graph attention network to obtain drug interaction representations, enabling the system to learn safe drug pair combinations. Moreover, to avoid overly conservative recommendations, we design a balanced loss function that weighs the benefits of a pair of drugs against the severity of any potential drug interaction. This approach allows for more effective and personalized treatment recommendations while minimizing the risk of adverse drug interactions. Additionally, since clinicians often titrate treatment regimes for patients based on their past lab test responses and medication dosages, we incorporate such trend information and adapt the vision transformer [7] (ViT) to learn an effective patient representation.

Our contributions can be summarized as follows: (1) We design a novel deep learning system that combines knowledge from patient information over multiple visits and severity-based drug interaction database to provide personalized, safe, and accurate fine-grained medication recommendation. (2) We propose to extract and model trend information from the medication dosage titrations and lab test responses over multiple visits to obtain effective patient representation. This enables the system to adjust the treatment regime as the patient condition changes. (3) We design a balanced DDI loss function that weighs the benefits of a pair of drugs against the severity of any potential drug interaction. (4) We provide comprehensive quantitative and qualitative evaluations on a benchmark medical dataset MIMIC-IV [10] and a proprietary outpatient dataset to demonstrate the capability and effectiveness of our proposed approach in providing fine-grained medication recommendations while considering drug interaction severity and patient history.

## 2   Related Work

Early works in medication recommendation focus on learning a collection of rules from EHR. Solt and Tikk [17] extract rules from discharge summaries, while Lakkaraju et al. [11] use Markov Decision Processes to learn the mapping between the patient characteristics and treatments. However, this approach may introduce conflicting rules, and is difficult to generalize and scale. Subsequent works employ recurrent neural networks to model sequential dependency in patient's past visits to improve the performance of medication recommendation [1, 12]. However, they do not consider drug interactions, which are crucial to minimize adverse drug reactions.

LEAP [27] incorporates an external drug-drug interaction (DDI) database and uses a recurrent decoder to model the drug-disease relationship in the current visit for recommending a set of medications. GameNet [16] uses dynamic memory and Graph Convolutional networks (GCN) to personalize the medication recommendations based on patients' longitudinal visit history and drug interactions. CompNet [21] is another GCN based recommender system that models relation among patient's past visits as well as the drug interactions by combining reinforcement learning with relational GCN.

PREMIER [2] introduces a two-stage recommender system that employs dual attention-based recurrent neural networks to model patient visits and adapts graph attention networks to model drug co-occurrences and drug interactions. SafeDrug [25] uses RNN based patient visit representation and incorporates drug molecular structure information to enhance the safety of the recommended drugs. They introduce a threshold during training to penalize the recommended set of medications if the total drug interaction in the set is above the threshold. MICRON [24] uses recurrent residual networks to capture the medication changes over the visits, and incorporate drug interaction information from an external database in the design of their DDI loss function.

COGNet [22] introduces a *copy* module, which generates the probability of a medication being repeated from past visits. They model the drug interaction information with the help of a GCN and a DDI loss function. MoleRec [26] introduces a hierarchical architecture to model the association between drug substructures at the molecular level and the target disease to provide personalized medication recommendation. [18] proposes a causal inference based medication recommendation model. They model patient visit information as causal graphs and DDI as propositional satisfiability

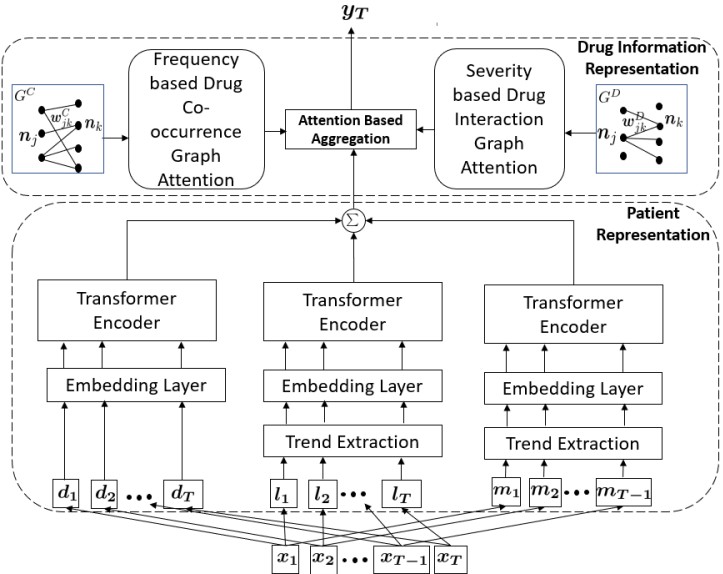

Figure 1: Overview of REFINE.

problem. However, they use additional information such as patient symptoms which are not available in our datasets, making a direct comparison difficult.

Our approach differs from existing works in that we provide fine-grained medication recommendation by explicitly modeling drug names, and the trend information in the medication dosages as well as the lab test responses gathered over multiple visits. We also ensure the safety of the recommendations by modeling the severity of drug interactions to reduce the risk of harmful combinations.

## 3 Methodology

Figure 1 shows the framework of the proposed system REFINE for REcommendation of FINE-grained medication. The input is a list of patient visits. The $i^{th}$ visit is represented as $\boldsymbol{x}_i = [\boldsymbol{d}_i, \boldsymbol{l}_i, \boldsymbol{m}_i]$ where $\boldsymbol{d}_i$ is a multi-hot vector of diagnosis, $\boldsymbol{l}_i$ is a vector of lab tests and responses, $\boldsymbol{m}_i$ is the vector of medications and dosages, and $i \in [1, T]$. All continuous values are normalized using min-max normalization [9]. The goal is to predict a set of medications $\boldsymbol{y_T}$ for the $T^{th}$ visit, taking into account the patient history, the trend in the medication dosages and lab test responses, as well as the severity of drug interactions.

### 3.1 Trend Information Extraction

We observe that the treatment regimes for patients with the same disease vary depending on whether their conditions are well managed. This is typically reflected in the fluctuations in the lab test responses over the various visits. Further, clinicians often titrate the dosage of first-line medications before introducing additional medications to achieve the optimal treatment outcome. As such, we extract trend information from medication dosages and lab test responses over the past visits in the form of slope and variance from [15].

Let $\mathbf{var}_i^m$ and $\mathbf{slope}_i^m$ be the vectors of dosage variance and rate of dosage change for the medications across visits 1 to $i$ respectively. The dosage variance and rate of dosage change for the $k^{th}$ medication can be computed as follows:

$$\text{var}_i^m[k] = \frac{\sum_{j=1}^{i}(m_j[k] - \mu_i^m[k])^2}{(i-1)} \tag{1}$$

$$\text{slope}_i^m[k] = \frac{\sum_{j=1}^{i}(t_j - \bar{t}_i)(m_j[k] - \mu_i^m[k])}{\sum_{j=1}^{i}(t_j - \bar{t}_i)^2} \tag{2}$$

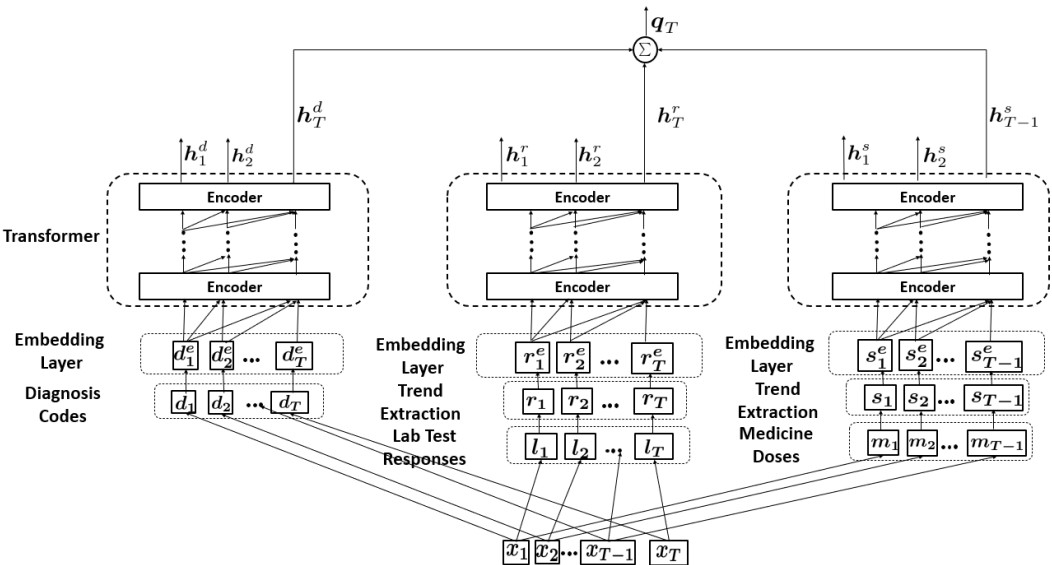

Figure 2: Patient representation obtained from transformer encoders.

where $m_j[k]$ is the dosage of the $k^{th}$ medication at the $j^{th}$ visit, $\mu_i^m[k]$ is the mean of the dosages of the $k^{th}$ medication across visits 1 to $i$, $t_j$ is the interval (in days) between the $1^{st}$ visit and the $j^{th}$ visit, $\bar{t}_i$ is the average $t_j$ for j $\in [1, i]$. The variance $\mathbf{var}_i^l$ and slope $\mathbf{slope}_i^l$ for the lab test responses are obtained in the similar manner.

We stack the variance $\mathbf{var}_i^m$ and slope $\mathbf{slope}_i^m$ and medication dosages $\boldsymbol{m}_i$ to obtain a 2D array $\in \mathbb{R}^{3 \times N_m}$ where $N_m$ is the total number of medications. We flatten the 2D array into a 1D array sequence, similar to the approach used in Vision Transformer (ViT) [7]. To do this, we concatenate the three values (medication dosage, slope, and variance) for every medication in a visit to obtain a 1D array sequence $\boldsymbol{s}_i$. We repeat the process for the lab test response information to obtain a 1D array sequence $\boldsymbol{r}_i$.

## 3.2   Patient Representation

To obtain a comprehensive representation of a patient's evolving health condition, it is crucial to capture and combine the sequential dependencies present in their EHR data. Specifically, we utilize three separate transformers [7, 20] to model the sequential dependencies in the patient's diagnosis, lab test response, and medication information over multiple visits. These transformer modules can be used individually or in combination to derive a patient representation even when some information sources are missing from the EHR data. The embeddings $\boldsymbol{d}_i^e$, $\boldsymbol{r}_i^e$, and $\boldsymbol{s}_i^e$ for diagnosis, lab test response, and medication dosage respectively are passed to their respective transformers to generate the patient representation $\boldsymbol{q}_T$ as shown in Figure 2.

Each transformer is composed of multiple layers of encoders, and each encoder has a multi-head self-attention block followed by a position-wise feed-forward network. For the $i^{th}$ visit, the encoder in the first layer of the transformer takes as input the concatenation of the respective input embeddings from visits 1 to $i$. The input to the subsequent encoders is the concatenation of the outputs from the encoder of the previous layer. This helps to preserve and process the information from the patient's visit history. Residual connections are employed around self-attention mechanism to facilitate the propagation of lower layer features to higher layers. To stabilize and accelerate neural network training, we apply layer normalization after the multi-head self-attention block and the position-wise feed-forward network. We also apply dropout to avoid over-fitting. The outputs of the transformers for the $i^{th}$ visit are $\boldsymbol{h}_i^d$, $\boldsymbol{h}_i^r$, and $\boldsymbol{h}_i^s$ for diagnosis, lab test response, and medication dosage respectively.

Since our goal is to predict the set of medications for the current visit $T$, we fuse the encoded sequential representation for the diagnosis and lab test response up to the current visit, and the encoded representation for the medication dosage information up to the previous visit to obtain the

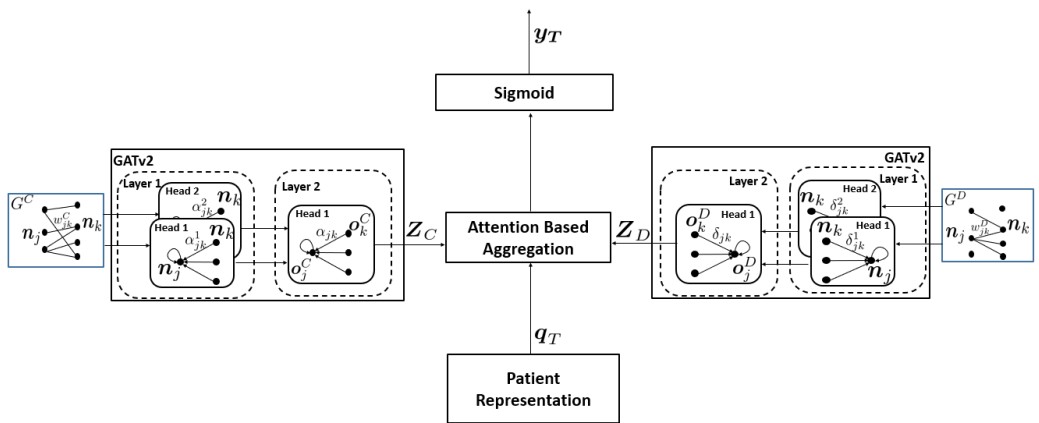

Figure 3: Details of modeling drug information from the EHR as well as the drug interaction database.

final patient representation $\boldsymbol{q}_T$ as follows:

$$\boldsymbol{q}_T = \boldsymbol{h}_T^d + \boldsymbol{h}_T^r + \boldsymbol{h}_{T-1}^s \tag{3}$$

## 3.3 Drug Information Representation

We observe that clinicians take into account the severity levels of interacting drug pairs and the severity of drug interactions can vary significantly. This information is crucial when prescribing medications to ensure patient safety. In addition, the frequency of co-occurrence of drug pairs observed in the EHR can provide valuable insights into commonly or rarely prescribed combinations, further guiding the decision-making process.

We represent drug co-occurrence information from the EHR and drug interaction from the DDI database DDInter [23] as two separate graphs $G^C = (V, E^C)$ and $G^D = (V, E^D)$ respectively. Each node $\boldsymbol{n}_i \in V$ depicts a drug $i$. The weighted edges in $E^C$ and $E^D$ depicts the co-occurrence frequency and interaction severity between drug pairs, respectively. For co-occurrence, a weighted edge $(\boldsymbol{n}_i, \boldsymbol{n}_j, w_{ij}^C) \in E^C$ gives the co-occurrence of drug $i$ and drug $j$ with a frequency of $w_{ij}^C$. The co-occurrence frequency is calculated by counting the number of occurrences of a drug pair in the prescribed medication sets in the EHR. These values are normalized to range between 0 and 1.

For drug interactions, a weighted edge $(\boldsymbol{n}_i, \boldsymbol{n}_j, w_{ij}^D) \in E^D$ gives the interaction between drug $i$ and drug $j$ with a severity of $w_{ij}^D$. The severity information is given as *Major*, *Moderate*, *Minor*, or *Unknown/No interaction* in the DDInter database. In this work, we represent these severity levels numerically as 1, 0.66, 0.33, and 0, respectively.

We adapt GATv2 [5] to learn node representations from the weighted drug interaction graph $G^D$. GATv2 has two layers with $z$ attention heads in the first layer, and 1 attention head in the second layer (see Figure 3). For the first layer, the attention weight between nodes $\boldsymbol{n}_j$ and $\boldsymbol{n}_k$ at the $b^{th}$ attention head is calculated as follows:

$$\delta_{jk}^b = \frac{w_{jk}^D \times \exp(\boldsymbol{a}'\text{LeakyReLU}(\boldsymbol{E}^b \cdot [\boldsymbol{n}_j \| \boldsymbol{n}_k]))}{\sum_{\boldsymbol{n}_i \in S_j} w_{ij}^D \times \exp(\boldsymbol{a}'\text{LeakyReLU}(\boldsymbol{E}^b \cdot [\boldsymbol{n}_j \| \boldsymbol{n}_i]))} \tag{4}$$

where $S_j$ is the set of drugs that interact with drug $j$, $a'$ is the transpose of the weight vector of a single layer feed-forward neural network, $\boldsymbol{E}^{b^*}$ is the embedding matrix for the $b^{th}$ attention head with $\boldsymbol{E}^b = [\boldsymbol{E}^{b^*} \| \boldsymbol{E}^{b^*}]$ [5], $w_{jk}^D$ is the edge weight depicting the severity of interaction between drugs $j$ and $k$, and $\|$ denotes the concatenation operation. The output from the first layer for node $n_j$ is given by:

$$\boldsymbol{o}_j^D = \|_{b=1}^z \sigma \left( \sum_{\boldsymbol{n}_k \in S_j} \delta_{jk}^b \boldsymbol{E}^{b^*} \cdot \boldsymbol{n}_k \right) \tag{5}$$

where $\sigma$ is the sigmoid function. With this, the output from the second layer for node $n_j$ is given by:

$$\boldsymbol{u}_j^D = \sigma \left( \sum_{\boldsymbol{n}_k \in \mathcal{S}_j} \delta_{jk} \boldsymbol{E} \cdot \boldsymbol{o}_k^D \right) \tag{6}$$

where $\boldsymbol{E}$ is the embedding matrix for the second layer, and $\delta_{jk}$ is obtained by applying Equation 4 on the output of the first layer for nodes $j$ and $k$ with $w_{jk}^D$ updated to the average of $\delta_{jk}^b$ for $1 \le b \le z$.

We obtain the drug interaction representation $\boldsymbol{Z}_D$ by stacking $\boldsymbol{u}_j^D$, $j \in \{1, 2, ..., N_m\}$ and $N_m$ is the total number of drugs. Similarly, we can obtain a drug co-occurrence representation $\boldsymbol{Z}_C$ by applying another GATv2 on the drug co-occurrence graph $G^C$ and stacking the node representation $\boldsymbol{u}_j^C$, $j \in \{1, 2, ..., N_m\}$. The final drug information representation $\boldsymbol{Z}$ is given by:

$$\boldsymbol{Z} = \boldsymbol{Z}_C + w\boldsymbol{Z}_D \tag{7}$$

where $w$ is a parameter that controls the relative importance of the two representations.

The final output vector $\boldsymbol{y_T}$ is obtained by applying attention-based aggregation on the patient representation and the weighted drug information representation as follows:

$$\boldsymbol{y_T} = \sigma(\boldsymbol{W} \cdot (\boldsymbol{q}_T + \boldsymbol{Z} \cdot \boldsymbol{\lambda})) \tag{8}$$

where $\boldsymbol{\lambda}$ = softmax( transpose($\boldsymbol{Z}$) $\cdot$ $\boldsymbol{q}_T$) and $\boldsymbol{W}$ is the gradient matrix of the transformation function in the linear layer.

We consider a medication is recommended if $y_T[k] > 0.5$, $k \in \{1, 2, ..., N_m\}$. To understand the factors influencing the model's recommendations, Appendix A shows how we can derive the contributions of the different information sources.

## 3.4 Training Objective

We formulate the recommendation task as a multi-label prediction task and employ the widely used binary cross entropy loss $L_{bce}$ and hinge loss $L_{hinge}$ [16, 3]. To capture the trade-off between the benefits and potential interactions of a drug pair, we propose a balanced drug interaction loss function $L_{bdi}$ defined as follows:

$$L_{bdi} = \sum_{i=1}^{T} \sum_{j=1}^{N_m} \sum_{k=1}^{N_m} (\beta w_{jk}^D - (1 - \beta) w_{jk}^C) \cdot y_i[j] \cdot y_i[k] \tag{9}$$

where $w_{jk}^D$ and $w_{jk}^C$ depict the severity of drug interaction and the frequency of co-occurrence obtained from the graphs $G^D$ and $G^C$ respectively, $\beta$ is a controllable factor which provides a balance between drug interaction and drug co-occurrence information.

The final objective function is given by:

$$L_{total} = \gamma_1 * L_{bce} + \gamma_2 * L_{hinge} + (1 - \gamma_1 - \gamma_2) * L_{bdi} \tag{10}$$

where $\gamma_1$ and $\gamma_1$ are hyperparameters.

## 4 Performance Study

We implement REFINE in PyTorch and carry out the training on two NVIDIA Titan RTX GPU. We use two datasets, MIMIC-IV [10] and PRIVATE, in our experiments. MIMIC-IV is a publicly available inpatient dataset consisting of 299,712 patient (mostly ICU patients) data collected between 2008 and 2019, whereas PRIVATE is a proprietary outpatient dataset from 6 primary care clinics over a span of ten years. Table 1 gives the statistics. Patients with fewer than two visits are filtered out (see details in Appendix B).

We evaluate the accuracy as well as the level of drug interactions in the recommended medication set. For accuracy, we use the metrics precision-recall area under the curve (AUC), F1, and Jaccard

Table 1: Characteristics of datasets.

| Attribute | MIMIC-IV | PRIVATE |
|---|---|---|
| Number of patients | 61310 | 85039 |
| Number of diagnosis | 2000 | 20 |
| Number of lab tests | 573 | 18 |
| Number of medications | 735 | 50 |
| Avg number of visits per patient | 4.0 | 6.9 |
| Avg number of diagnosis per visit | 12.5 | 1.8 |
| Avg number of lab tests per visit | 35.6 | 5.6 |
| Avg number of medications per visit | 8.9 | 4.1 |

Table 2: Results of comparative study.

| Methods | MIMIC-IV | | | PRIVATE | | |
|---|---|---|---|---|---|---|
| | AUC | F1 | wDDI | AUC | F1 | wDDI |
| LR | $0.632 \pm 0.006$ | $0.526 \pm 0.004$ | $0.055 \pm 0.0009$ | $0.639 \pm 0.007$ | $0.561 \pm 0.002$ | $0.146 \pm 0.0004$ |
| RF | $0.635 \pm 0.004$ | $0.531 \pm 0.003$ | $0.053 \pm 0.0005$ | $0.642 \pm 0.003$ | $0.565 \pm 0.005$ | $0.140 \pm 0.0003$ |
| BART | $0.637 \pm 0.002$ | $0.540 \pm 0.002$ | $0.056 \pm 0.0006$ | $0.646 \pm 0.004$ | $0.571 \pm 0.004$ | $0.142 \pm 0.0005$ |
| LEAP | $0.625 \pm 0.003$ | $0.514 \pm 0.005$ | $0.047 \pm 0.0003$ | $0.633 \pm 0.005$ | $0.558 \pm 0.007$ | $0.117 \pm 0.0001$ |
| DMNC | $0.633 \pm 0.006$ | $0.528 \pm 0.002$ | $0.056 \pm 0.0002$ | $0.640 \pm 0.007$ | $0.566 \pm 0.002$ | $0.158 \pm 0.0005$ |
| GameNet | $0.641 \pm 0.003$ | $0.543 \pm 0.005$ | $0.053 \pm 0.0005$ | $0.647 \pm 0.001$ | $0.575 \pm 0.004$ | $0.131 \pm 0.0005$ |
| CompNet | $0.638 \pm 0.002$ | $0.546 \pm 0.006$ | $0.050 \pm 0.0002$ | $0.645 \pm 0.005$ | $0.582 \pm 0.001$ | $0.134 \pm 0.0005$ |
| PREMIER | $0.655 \pm 0.001$ | $0.565 \pm 0.002$ | $0.047 \pm 0.0001$ | $0.669 \pm 0.001$ | $0.607 \pm 0.003$ | $0.122 \pm 0.0001$ |
| MICRON | $0.658 \pm 0.003$ | $0.567 \pm 0.006$ | $0.047 \pm 0.0006$ | $0.662 \pm 0.002$ | $0.597 \pm 0.006$ | $0.120 \pm 0.0001$ |
| SafeDrug | $0.652 \pm 0.004$ | $0.564 \pm 0.002$ | $0.045 \pm 0.0001$ | $0.671 \pm 0.003$ | $0.608 \pm 0.006$ | $0.113 \pm 0.0002$ |
| COGNet | $0.661 \pm 0.002$ | $0.573 \pm 0.001$ | $0.054 \pm 0.0006$ | $0.682 \pm 0.005$ | $0.616 \pm 0.002$ | $0.129 \pm 0.0004$ |
| MoleRec | $0.670 \pm 0.004$ | $0.579 \pm 0.003$ | $0.044 \pm 0.0001$ | $0.688 \pm 0.002$ | $0.623 \pm 0.006$ | $0.112 \pm 0.0001$ |
| REFINE | $\mathbf{0.708 \pm 0.001}$* | $\mathbf{0.635 \pm 0.003}$* | $\mathbf{0.043 \pm 0.0001}$* | $\mathbf{0.729 \pm 0.004}$* | $\mathbf{0.656 \pm 0.003}$* | $\mathbf{0.111 \pm 0.0002}$* |

\* indicates that the result is statistically significant when compared to the second best with p-value $< 0.05$.

similarity [2, 25]. For the level of drug interactions, we introduce a weighted drug interaction score (wDDI) defined as follows:

$$\text{wDDI} = \frac{1}{T} \sum_{i=1}^{T} \left( \frac{2}{|M_i||M_i - 1|} \sum_{j \in M_i} \sum_{k \neq j \in M_i} w_{jk}^D \right) \tag{11}$$

where $M_i$ is the set of recommended medications at the $i^{th}$ visit for a patient, $|M_i|$ is the cardinality of $M_i$, $w_{jk}^D$ is the severity of interaction between drug $j$ and drug $k$ obtained from the graph $G^D$. The values of the various metrics are averaged over all the patients in the test set.

We split the datasets into training, validation and test sets in the ratio 4:1:1, and report the performance on the test set. Appendix C provides details of the training and hyperparameter settings. Sensitivity experiments and additional comparative and ablation studies can be found in Appendix D.

### 4.1 Comparative Analysis

We compare the performance REFINE with baseline methods such as Logistic Regression (LR) [13], Random Forest (RF) [4], BART [6] LEAP [27], DMNC [12], GameNet [16], CompNet [21], PRE-MIER [2], MICRON [24], SafeDrug [25], COGNet [22], and MoleRec [26]. We perform 10 rounds of bootstrapping sampling and report the mean and standard deviation.

As can be seen from Table 2, REFINE demonstrates superior performance compared to the baseline methods, with statistically significant differences (p-values $< 0.05$) based on a one-way ANOVA test [8]. The improved performance of REFINE can be attributed to its ability to account for medication dosage and lab test response trends across patient visits. Further, REFINE also outperforms the baselines in terms of wDDI, and is even lower than the wDDI present in MIMIC-IV (0.0448) and PRIVATE (0.1159) datasets. This suggests that REFINE can effectively reduce the potential risk of adverse drug interactions, which is a critical aspect of ensuring patient safety in clinical practice.

Table 3 shows the performance on a subset of patient instances where at least one change of medication occurs between consecutive visits. As more than 90% of the instances in MIMIC-IV already involve medication changes, we consider the performance reported in Table 2 to be representative of such

Table 3: Results for instances with medication change.

| Methods | PRIVATE | | | |
|---|---|---|---|---|
| | AUC | F1 | Jaccard | wDDI |
| LR | $0.605 \pm 0.006$ | $0.536 \pm 0.004$ | $0.379 \pm 0.008$ | $0.1239 \pm 0.0006$ |
| RF | $0.607 \pm 0.003$ | $0.539 \pm 0.005$ | $0.381 \pm 0.005$ | $0.1234 \pm 0.0002$ |
| BART | $0.609 \pm 0.002$ | $0.541 \pm 0.003$ | $0.385 \pm 0.004$ | $0.1236 \pm 0.0004$ |
| LEAP | $0.602 \pm 0.003$ | $0.527 \pm 0.005$ | $0.374 \pm 0.006$ | $0.1111 \pm 0.0002$ |
| DMNC | $0.606 \pm 0.006$ | $0.532 \pm 0.002$ | $0.387 \pm 0.005$ | $0.1243 \pm 0.0003$ |
| GameNet | $0.619 \pm 0.003$ | $0.545 \pm 0.005$ | $0.392 \pm 0.003$ | $0.1124 \pm 0.0006$ |
| CompNet | $0.614 \pm 0.002$ | $0.552 \pm 0.006$ | $0.401 \pm 0.008$ | $0.1138 \pm 0.0003$ |
| PREMIER | $0.638 \pm 0.001$ | $0.576 \pm 0.002$ | $0.419 \pm 0.005$ | $0.1119 \pm 0.0001$ |
| MICRON | $0.647 \pm 0.003$ | $0.581 \pm 0.006$ | $0.422 \pm 0.002$ | $0.1114 \pm 0.0004$ |
| SafeDrug | $0.648 \pm 0.004$ | $0.591 \pm 0.002$ | $0.405 \pm 0.006$ | $0.1107 \pm 0.0001$ |
| COGNet | $0.665 \pm 0.007$ | $0.605 \pm 0.002$ | $0.437 \pm 0.005$ | $0.1129 \pm 0.0007$ |
| MoleRec | $0.667 \pm 0.003$ | $0.610 \pm 0.001$ | $0.442 \pm 0.003$ | $0.1106 \pm 0.0002$ |
| **REFINE** | $\mathbf{0.707 \pm 0.002}$* | $\mathbf{0.645 \pm 0.001}$* | $\mathbf{0.494 \pm 0.003}$* | $\mathbf{0.1097 \pm 0.0001}$* |

\* indicates that the result is statistically significant when compared to the second best with p-value $< 0.05$.

Table 4: Average number of recommended, extra, and missed medications.

| Methods | MIMIC-IV | | | PRIVATE | | |
|---|---|---|---|---|---|---|
| | # Average | # Extra | # Missed | # Average | # Extra | # Missed |
| PREMIER | 9.85 | 4.60 | 3.47 | 5.13 | 2.3 | 1.35 |
| MICRON | 7.50 | 2.90 | 4.12 | 3.08 | 0.91 | 2.01 |
| SafeDrug | 7.01 | 2.57 | 4.29 | 2.93 | 0.77 | 2.02 |
| COGNet | 10.31 | 4.86 | 3.27 | 5.27 | 2.36 | 1.27 |
| MoleRec | 10.27 | 4.77 | 3.22 | 5.22 | 2.29 | 1.25 |
| REFINE | 9.48 | 3.70 | 2.94 | 4.91 | 1.93 | 1.20 |

cases. Therefore, we focus our analysis on the PRIVATE dataset, where around 50% of the instances involve medication changes between visits. We observe a drop in performance for all methods. This is expected as a change in treatment regime often implies a different patient response, making accurate predictions more challenging. Nevertheless, REFINE still outperforms the baseline methods in this challenging scenario. This indicates that even when faced with more complex situations involving medication changes, REFINE's integrated approach provides valuable insights and maintains a higher level of performance compared to other methods.

We also analyze the average number of recommended medications, average number of extra medications recommended, and average number of missed medications per visit for REFINE and the best performing baselines, as shown in Table 4. The average number of medications in the ground truth for the test set is 8.72 and 4.18 for MIMIC-IV and PRIVATE dataset respectively. We see that COGNet, PREMIER, and MoleRec generally recommend more medications whereas SafeDrug, MICRON tend to recommend fewer medications; whereas the average number of medications recommended by REFINE is closely aligned with the ground truth for both datasets. We find that COGNet has the highest average of extra medications in the recommended set. In contrast, REFINE recommends relatively lower extra medications. This can be attributed to the effective learning of patient condition from lab test and medication dosage trends. For the average number of missed medications, SafeDrug has the worst performance. On the contrary, REFINE misses the least number of medications from the ground truth set. This indicates that not weighing the benefits and potential interactions of drug pairs together may lead the system to omit medications necessary for managing a patient's condition.

## 4.2 Ablation Study

We conduct an ablation study to evaluate the effect of trend information and drug interaction severity on the performance of REFINE. We have four variants: (a) **REFINE w/o med trend** excludes variance and slope of the medication dosages, (b) **REFINE w/o lab trend** omits the variance and slope of the lab test responses, (c) **REFINE w/o any trends** does not incorporate any trend information, and (d) **REFINE w/o severity** does not incorporate severity-based drug interaction.

Table 5 shows the results. We observe that **REFINE w/o any trend** has the lowest accuracy, indicating that trend information is crucial for making accurate fine-grained medication recommendations. There is a significant drop in performance for **REFINE w/o severity**, highlighting the importance of considering drug interaction severity when recommending medications.

Table 5: Results for ablation study.

| Methods | MIMIC-IV | | | PRIVATE | | |
|---|---|---|---|---|---|---|
| | AUC | F1 | wDDI | AUC | F1 | wDDI |
| w/o med trend | $0.701 \pm 0.001$ | $0.618 \pm 0.003$ | $0.044 \pm 0.0001$ | $0.718 \pm 0.003$ | $0.645 \pm 0.004$ | $0.112 \pm 0.0002$ |
| w/o lab trend | $0.695 \pm 0.002$ | $0.613 \pm 0.001$ | $0.044 \pm 0.0003$ | $0.714 \pm 0.004$ | $0.643 \pm 0.001$ | $0.112 \pm 0.0005$ |
| w/o any trends | $0.682 \pm 0.002$ | $0.602 \pm 0.002$ | $0.045 \pm 0.0002$ | $0.703 \pm 0.002$ | $0.630 \pm 0.001$ | $0.112 \pm 0.0004$ |
| w/o severity | $0.694 \pm 0.002$ | $0.617 \pm 0.001$ | $0.046 \pm 0.0005$ | $0.713 \pm 0.004$ | $0.639 \pm 0.002$ | $0.115 \pm 0.0003$ |
| **REFINE** | $\mathbf{0.708 \pm 0.001}$ * | $\mathbf{0.635 \pm 0.003}$* | $\mathbf{0.043 \pm 0.0001}$* | $\mathbf{0.729 \pm 0.004}$ * | $\mathbf{0.656 \pm 0.003}$* | $\mathbf{0.111 \pm 0.0002}$* |

\* indicates that the result is statistically significant when compared to the second best with p-value $< 0.05$.

## 5  Case Studies

Finally, we present case studies to highlight REFINE's ability to provide accurate medication recommendations which ultimately contributes to better patient care. Figure 4(a) shows a sample patient from the PRIVATE dataset who has been diagnosed with Type 2 Diabetes Mellitus and Hyperlipidemia, and the HbA1c ranges from 7.4 to 9.8. In the second visit, we observe that a diabetes medication Glipizide is added to the Metformin treatment to manage the increased HbA1c value. REFINE successfully recommends the addition of Glipizide, while SafeDrug and COGNet do not. This can be attributed to REFINE's ability to learn lab test trends and incorporate balanced drug interaction information. REFINE identifies the increase in HbA1c value for the current visit as the influencing factor for Glipizide, which is consistent with clinical knowledge. Further, the decrease in HbA1c in the third visit indicates that certain medication combinations with some risk of interaction may indeed be necessary to effectively manage a patient's condition.

Figure 4(b) shows another patient from the PRIVATE dataset who has been diagnosed with Diabetes Mellitus, Hypertension, and Hyperlipidemia in visit 1, followed by chronic kidney disease in visit 2. We see that the medication Simvastatin is replaced by Atorvastatin in the second visit as it is preferred over other hyperlipidemia medications for patients with chronic kidney disease [2]. REFINE closely follows the prescribed medication set and accurately recommends the medication change, indicating REFINE's ability to model co-morbidities. In contrast, SafeDrug and COGNet are unable to effectively capture this aspect.

Figure 4(c) shows a patient from the MIMIC-IV dataset who has been diagnosed with multiple diseases in visit 1, and is diagnosed with nontoxic multinodular goiter, and nausea with vomiting in visit 2. We observe that REFINE is able to recommend medications that matches the prescribed medications more closely compared to COGNet and SafeDrug. A closer inspection reveals that SafeDrug tends to under-recommend to avoid the interacting medication pairs, whereas COGNet tends to over-recommend many drugs outside the prescribed set.

## 6  Conclusion and Discussion

We have described a unique fine-grained medication recommender system that offers personalized medication suggestions while considering the severity of potential drug interactions and explicitly modeling the trends in medication dosage titrations and lab test responses over visits. To the best of our knowledge, REFINE is the only system that factors in these trends while also assessing the varying severity of drug interactions. Experiments on real-world datasets and case studies show the effectiveness of REFINE in providing personalized, accurate, and safe medication recommendation.

The current system can be extended to include additional rich data from EHR such as clinician notes to further refine the system's recommendations. While we have shown that the severity of drug interactions is a crucial factor, the dosage strength of the drugs could also have a significant impact. A lower dosage of an interacting drug pair may sometimes be clinically acceptable and beneficial. Future work could explore how dosage strength influences the acceptability of interacting drug pairs.

## Acknowledgements

This research/project is supported by the National Research Foundation, Singapore under its AI Singapore Programme (AISG Award No: AISG-GC-2019-001-2A).

---

[2]https://tinyurl.com/3deyr6bp

|  | | Visit 1 | Visit 2 | Visit 3 |
|---|---|---|---|---|
| Diagnosis | | Diabetes mellitus, Hyperlipidemia | Diabetes mellitus, Hyperlipidemia | Diabetes mellitus, Hyperlipidemia |
| Lab Tests | | HbA1c | Cholesterol HDL, Cholesterol LDL, Cholesterol Total, HbA1c ↑, Triglycerides | ALT test, Cholesterol HDL, Cholesterol LDL, Cholesterol Total, HbA1c ↓, Triglycerides |
| Prescribed Medications | | Metformin , Simvastatin | Metformin, Glipizide, Simvastatin | Metformin ↓, Glipizide, Simvastatin |
| Recommended Medications | SafeDrug | Metformin , Simvastatin | Metformin , Atorvastatin | Metformin , Simvastatin |
| | COGNet | Metformin , Simvastatin | Metformin , Simvastatin, Atorvastatin | Metformin , Simvastatin |
| | REFINE | Metformin , Simvastatin | Metformin, Glipizide, Simvastatin | Metformin, Glipizide, Simvastatin |
| [Recommended Medications, Influencing Factors] | | [Metformin, Diabetes mellitus] , [Simvastatin, Hyperlipidemia] | [Metformin, Diabetes mellitus], [Glipizide, HbA1c ↑ ], [Simvastatin, Hyperlipidemia] | [Metformin, HbA1c ↓], [Glipizide, HbA1c ↓], [Simvastatin, Cholesterol LDL Calculated] |

(a) Patient with fluctuating HbA1c values.

|  | | Visit 1 | Visit 2 |
|---|---|---|---|
| Diagnosis | | Diabetes mellitus, Hypertension, Hyperlipidemia | Diabetes mellitus, Hypertension, Hyperlipidemia, Diabetic chronic kidney disease |
| Lab Tests | | ALT test, Chloride serum, Cholesterol HDL, Cholesterol LDL, Cholesterol Total, Creatinine, Glucose (fasting) plasma, HbA1c, Potassium, Sodium, Triglycerides | HbA1c ↓ |
| Prescribed Medications | | Metformin ,Glipizide, Simvastatin, Lisinopril | Metformin, Glipizide, Atorvastatin, Lisinopril |
| Recommended Medications | SafeDrug | Metformin , Simvastatin, Atenolol | Metformin , Simvastatin, Atenolol, Lisinopril |
| | COGNet | Metformin, Simvastatin, Enalapril | Metformin , Simvastatin, Atenolol, Lisinopril |
| | REFINE | Metformin, Glipizide, Simvastatin, Atenolol | Metformin, Glipizide, Atorvastatin, Lisinopril |
| [Recommended Medications, Influencing Factors] | | [Metformin, Diabetes mellitus], [Glipizide, HbA1c], [Simvastatin, Hyperlipidemia], [Atenolol, Hypertension] | [Metformin, Diabetes mellitus], [Glipizide, Diabetes mellitus], [Atorvastatin, Hyperlipidemia], [Lisinopril, Hypertension] |

(b) Patient with changes in co-morbidities.

|  | | Visit 1 | Visit 2 |
|---|---|---|---|
| Diagnosis | | Morbid (severe) obesity, Obstructive sleep apnea, Prediabetes, Postprocedural hypothyroidism, Vitamin D deficiency, Iron deficiency anemia, Hyperlipidemia, Hyperuricemia, Localized edema, Nausea | Nontoxic multinodular goiter, Nausea with vomiting |
| Lab Tests | | Hematocrit | Parathyroid Hormone, Calcium |
| Prescribed Medications | | Heparin, Lactate, Acetaminophen, Cefazolin, Levothyroxine, Famotidine | Heparin, Calcium carbonate, Hydromorphone, Calcitriol, Levothyroxine ↓, Scopolamine |
| Recommended Medications | SafeDrug | Heparin, Lactate, Acetaminophen, Levothyroxine, Simvastatin | Heparin, Levothyroxine, Scopolamine, Lactate |
| | COGNet | Lactate, Ibuprofen, Cefazolin, Levothyroxine, Famotidine, Simvastatin | Heparin, Calcium carbonate, Hydromorphone, , Levothyroxine, Famotidine, Lactate |
| | REFINE | Heparin, Lactate, Acetaminophen, Cefazolin, Levothyroxine, Famotidine, Simvastatin | Heparin, Calcium carbonate, Hydromorphone, Levothyroxine, Scopolamine |
| [Recommended Medications, Influencing Factors] | | [Heparin, Postprocedural hypothyroidism], [Lactate, Nausea], [Acetaminophen, Postprocedural hypothyroidism], [Cefazolin, Postprocedural hypothyroidism], [Levothyroxine, Postprocedural hypothyroidism], [Famotidine, Nausea], , [Simvastatin, Hyperlipidemia] | [Heparin, Postprocedural hypothyroidism], [Calcium carbonate, Calcium], [Hydromorphone, Postprocedural hypothyroidism], [Levothyroxine, Nontoxic multinodular goiter], [Scopolamine, Nausea with vomiting] |

(c) Patient from MIMIC-IV with multiple diagnosis.

Figure 4: Medications recommendeded for sample patients. Red color indicates interacting medications. Symbols ↑ and ↓ depict an increase and decrease in the value of dosage or lab test response. The top influencing factor for each recommended medication provide insights into why REFINE recommends it.

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
