# Appendix of REFINE: A Fine-Grained Medication Recommendation System Using Deep Learning and Personalized Drug Interaction Modeling

**Suman Bhoi**[1]     **Mong Li Lee**[1]     **Wynne Hsu**[1]     **Ngiap Chuan Tan**[2]

[1]National University of Singapore     [2]SingHealth Polyclinics
sumanbhoi@u.nus.edu, {leeml,whsu}@comp.nus.edu.sg, tan.ngiap.chuan@singhealth.com.sg

## A   Identification of Influencing Factors

In order to understand the factors that influence the recommendation provided by REFINE we calculate the contribution scores of the input information sources. Let us look into the factors influencing the recommendation of the $j^{th}$ medication for the current visit $T$ of a patient. From Equation 8 (in main paper) we can obtain the contribution score of each diagnosis as follows:

$$\boldsymbol{\eta}^d = \|_{i=1}^T a_T^d[i] \times (\boldsymbol{W}^y[j,:] \cdot \boldsymbol{E}^d[:,k]) \times d_i[k] \tag{1}$$

where $d_i[k]$ denotes the $k^{th}$ diagnosis in the $i^{th}$ visit, $\boldsymbol{E}^d$ is the embedding matrix for diagnosis, $\boldsymbol{W}^y$ is the gradient matrix of the final linear layer, $\boldsymbol{a}_T^d$ is the average of attention weights across all the heads and layers of the encoders in the transformer for the diagnosis information for time $T$, $a_T^d[i]$ is the $i^{th}$ entry in $\boldsymbol{a}_T^d$.

The contribution score of each lab test is given by:

$$\boldsymbol{\eta}^r = \|_{i=1}^T a_T^r[i] \times (\boldsymbol{W}^y[j,:] \cdot \boldsymbol{E}^r[:,k]) \times r_i[k] \tag{2}$$

where $r_i[k]$ denotes the $k^{th}$ lab test information in the $i^{th}$ visit, $\boldsymbol{E}^r$ is the embedding matrix for lab test, $\boldsymbol{a}_T^r$ is the average of attention weights across all the heads and layers of the encoders in the transformer for the lab test response information for time $T$, $a_T^r[i]$ is the $i^{th}$ entry in $\boldsymbol{a}_T^r$.

Similarly, the contribution score of the past prescribed medication is given by:

$$\boldsymbol{\eta}^s = \|_{i=1}^{T-1} a_{T-1}^s[i] \times (\boldsymbol{W}^y[j,:] \cdot \boldsymbol{E}^s[:,k]) \times s_i[k] \tag{3}$$

where $s_i[k]$ denotes the $k^{th}$ medication information in the $i^{th}$ visit, $\boldsymbol{E}^s$ is the embedding matrix for medication, $\boldsymbol{a}_{T-1}^s$ is the average of attention weights across all the heads and layers of the encoders in the transformer for the medication information for time $T-1$, $a_{T-1}^s[i]$ is the $i^{th}$ entry in $\boldsymbol{a}_{T-1}^s$.

Finally, the contribution score of drug co-occurrences and drug interactions are given by:

$$\boldsymbol{\eta}^C = \|_{k \in \mathcal{S}_j} \lambda[k] \times a_j^C[k] \times \boldsymbol{W}^y[j,:] \cdot \left( \boldsymbol{E}^C \cdot \boldsymbol{n}_k \right) \tag{4}$$

$$\boldsymbol{\eta}^D = \|_{k \in \mathcal{S}_j} w \times \lambda[k] \times a_j^D[k] \times \boldsymbol{W}^y[j,:] \cdot \left( \boldsymbol{E}^D \cdot \boldsymbol{n}_k \right) \tag{5}$$

where $\boldsymbol{n}_k$ is the one-hot vector representation of the $k^{th}$ node in the graphs, $\boldsymbol{E}^C$ and $\boldsymbol{E}^D$ are the average of embedding matrices $\boldsymbol{E}^b$ over all heads in GATv2 for $G^C$ and $G^D$ respectively, $\boldsymbol{a}_j^C$

37th Conference on Neural Information Processing Systems (NeurIPS 2023).

and $\boldsymbol{a}_j^D$ are the averages of attention weights across all heads and layers of GATv2 for the drug information graphs $G^C$ and $G^D$ respectively for node $j$, $a_j^C[k]$ and $a_j^D[k]$ are the $k^{th}$ entry in $\boldsymbol{a}_j^C$ and $\boldsymbol{a}_j^D$ respectively, $\lambda[k]$ is the attention weight for the $k^{th}$ node as described in Section 3.3.

We normalize the contribution scores and rank them to obtain the top factors that may have influenced the recommendation.

## B  Data Preprocessing

The MIMIC-IV and PRIVATE medical datasets use both ICD-9 and ICD-10 codes for diagnoses. Therefore, to ensure consistency we convert all the diagnosis information based on ICD-10 encoding. We use the medication name directly from the PRIVATE dataset. In contrast, the medication names data present in MIMIC-IV are noisy and hence we use the NDC codes to obtain the medication names. Specifically, we convert the NDC codes to RxNorm and then from RxNorm to medication generic name. All dosage values are converted to either milligrams or milliliters to ensure uniformity. Visits are identified based on unique hospital admission ID. Similar to the existing baselines, we consider the medications prescribed in the first 24 hours. We exclude patients with less than two visits. These pre-processing steps are crucial for eliminating noise and redundancy to ensure the accuracy and reliability of the datasets. We present the distribution of the number of visits in both datasets in Figure 1.

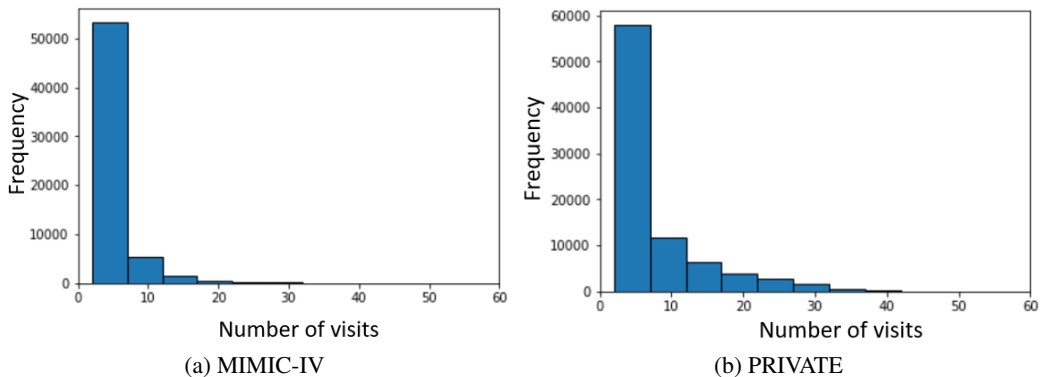

|  (a) MIMIC-IV | (b) PRIVATE |

Figure 1: Histogram depicting the distribution of visits in both datasets.

## C  Training and Hyperparameter Setting

**Training.** Algorithm 1 outline the steps involved in the training of REFINE. All the equations mentioned in the Algorithm refer to equations in the main paper. We split the datasets in to training, testing, and validation sets in the ratio 4:1:1. The performance of all the techniques is reported on the test set. The model has a fixed embedding size of 64, which is the dimensionality of the vector used to represent the input. During training, the Adam optimizer [2] is used with a learning rate of 0.0001. For the baseline techniques, we provide diagnosis, medication, and lab test items as input from the EHR. However, we do not incorporate the medication dosage information and the lab test response information for the baselines. Also, for baselines using DDI we use DDInter database [4] with the assumption that drug pairs with minor, moderate, and major interaction have the same weight of value 1. The best performing model is selected based on its performance on the validation set after 40 epochs.

**Hyperparameter settings.** For MIMIC-IV, REFINE uses a dropout of 0.4 on the input embedding layer, with a transformer encoder consisting of four layers and 2 attention heads each. We model the drug co-occurrence information and drug interaction information using two graph attention networks (GATv2). Each GATv2 consists of two layers with the first layer consisting of 4 attention heads and the second layer consisting of 1 attention head. The GATv2 parameter value is set to $w = 0.45$ based on the validation set. The controllable factor $\beta$ in L$_{bdi}$ is set to 0.65. For all the Recurrent Neural

---
**Algorithm 1** Training steps of REFINE
---
**Input**: Training set $\mathcal{X}$, training epochs N, Drug co-occurrence graph $G^C$, Drug interaction graph $G^D$, Loss weights $\gamma_1$ and $\gamma_2$, $w$ in Eq.(7), $\beta$ in Eq.(9).
**Parameter**: Learnable parameters of REFINE
**Output**: Set of recommended medications $\boldsymbol{y}$

  1: Obtain $\boldsymbol{Z}$ by using Eq.(4)-(7).
  2: **for** epoch = 1 to N **do**
  3:    **for** j = 1 to $|\mathcal{X}|$ **do**
  4:       Sample a patient $\boldsymbol{X}^{(j)} = [\boldsymbol{x}_1, \boldsymbol{x}_2, ..., \boldsymbol{x}_{T_j}]$
  5:       **for** i = 1 to $T_j$ **do**
  6:          Generate $\textbf{var}_i^m$, $\textbf{slope}_i^m$, $\textbf{var}_i^l$, $\textbf{slope}_i^l$ using Eq.(1) and (2) for patient $j$
  7:          Obtain $\boldsymbol{h}_i^d$, $\boldsymbol{h}_i^r$, and $\boldsymbol{h}_{i-1}^s$ from transformer encoders and generate $\boldsymbol{q}_i$ using Eq.(3)
  8:          Obtain $\boldsymbol{y}_i$ from Eq.(8)
  9:       **end for**
10:       Update parameters by optimizing $L_{\text{total}}$ in Eq.(10)
11:    **end for**
12: **end for**
---

Network based techniques, a GRU with a hidden dimension of 64 is used. The hyperparameters for all the baselines were chosen on the validation set. The DMNC model uses a word size of 64 and a memory size of 16, following the original work. We set the maximum drug combination size at 40 for both LEAP and DMNC. For GAMENet, we follow a DDI rate of 0.05, weight decay of 0.86, and mixture weights $\pi = [0.9, 0.1]$. For CompNet, we use 3 convolutional layer with filter sizes set to 128 in the patient representation module. For PREMIER, we set $\gamma_1 = 0.80$ and $\gamma_2 = 0.02$ with a dropout rate of 0.4 on the input embedding layer. For MICRON, the best performance is reported for visit weight $\gamma = 0.78$, loss weights $\lambda_i = 0.25$, $i = 1,2,3,4$ and DDI threshold $\eta = 0.07$. For SafeDrug, we set the loss weight $\alpha = 0.95$, correcting factor $K_p = 0.05$, and acceptance rate $\gamma = 0.07$. For COGNet, we set the number of beam search state at 4 and the maximum generation length at 40. For MoleRec, the hyperparameters $\delta$, $\beta$, $\tau$, $\phi$ are set at 0.50, 0.90, 0.25, and 0.08 respectively.

For PRIVATE, REFINE uses a dropout of 0.5 on the input embedding layer, with a transformer encoder consisting of four layers with 2 attention heads each. The drug co-occurrence information and drug interaction information is modeled using two graph attention networks (GATv2) with two layers consisting of 2 attention heads in the first layer, and 1 attention head in the second layer. The GATv2 parameter value is set to $w = 0.40$. The controllable factor $\beta$ in $L_{\text{bdi}}$ is set to 0.55. For all the Recurrent Neural Network based techniques, a GRU with a hidden dimension of 64 is used. The hyperparameters for all the baselines were chosen on the validation set. The DMNC model uses a word size of 64 and a memory size of 32. We set the maximum drug combination size at 30 for both LEAP and DMNC. For GAMENet, we follow a DDI rate of 0.10, weight decay of 0.86, and mixture weights $\pi = [0.85, 0.15]$. For CompNet, we use 3 convolutional layer with filter sizes set to 128 in the patient representation module. For PREMIER, we set $\gamma_1 = 0.78$ and $\gamma_2 = 0.05$ with a dropout rate of 0.5 on the input embedding layer. For MICRON, the best performance is reported for visit weight $\gamma = 0.75$, loss weights $\lambda_i = 0.25$, $i = 1,2,3,4$ and DDI threshold $\eta = 0.10$. For SafeDrug, we set the loss weight $\alpha = 0.95$, correcting factor $K_p = 0.05$, and acceptance rate $\gamma = 0.10$. For COGNet, we set the number of beam search state at 4 and the maximum generation length at 25. For MoleRec, the hyperparameters $\delta$, $\beta$, $\tau$, $\phi$ are set at 0.50, 0.95, 0.22, and 0.13 respectively.

# D   Additional Experiments

## D.1   Sensitivity Experiments

We examine the impact of input embedding dimension, number of heads per layer in GATv2, number of layers and number of attention heads per layer in the transformer encoders as well as the loss weights $\gamma_1$, $\gamma_2$ on the performance of REFINE.

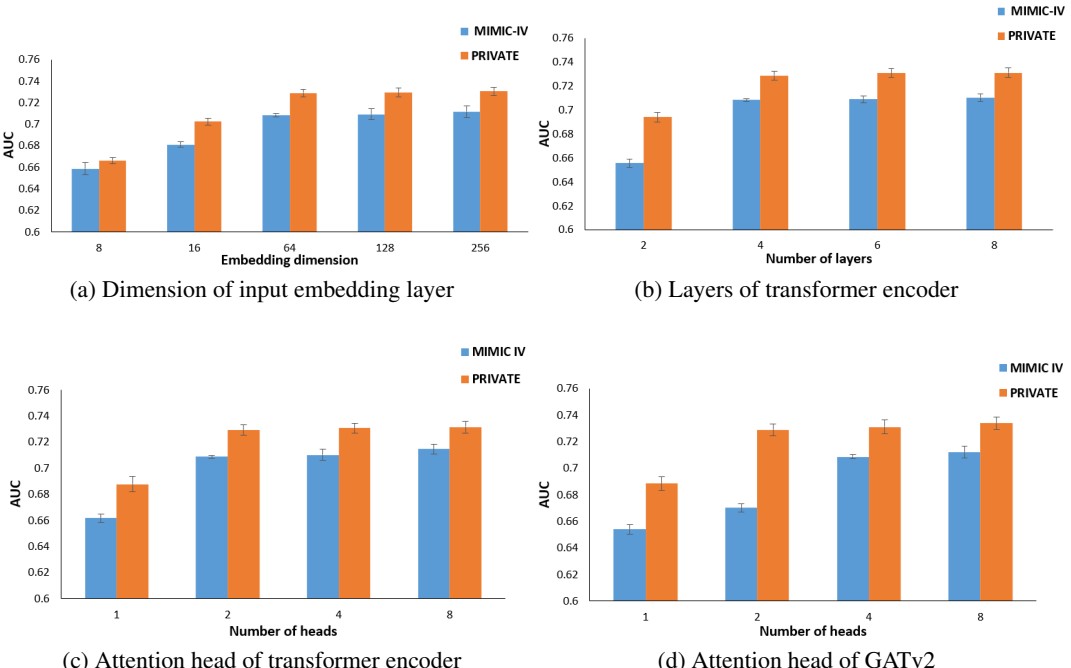

(a) Dimension of input embedding layer

(b) Layers of transformer encoder

(c) Attention head of transformer encoder

(d) Attention head of GATv2

Figure 2: Effect of embedding dimension and attention heads on REFINE.

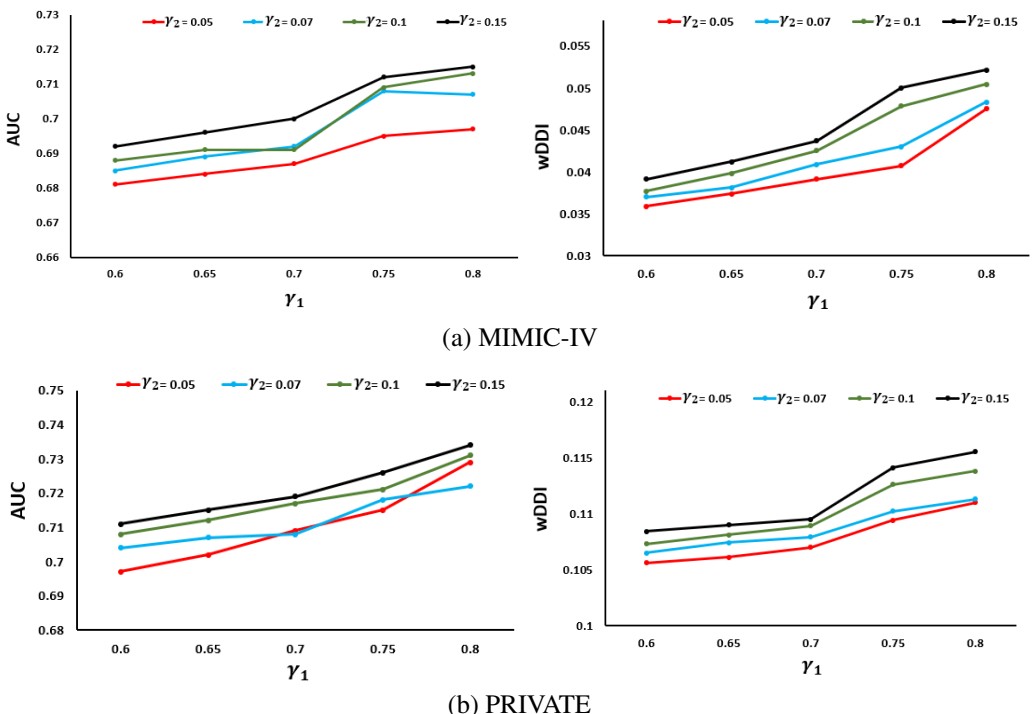

(a) MIMIC-IV

(b) PRIVATE

Figure 3: Effect of loss weights on REFINE.

Figure 2(a) depicts the impact of varying input embedding dimension on the performance of REFINE. We observe that the embedding dimension of 64 provides the optimal AUC for both MIMIC-IV and PRIVATE datasets.

Table 1: Results for comparative study.

| Methods | MIMIC-IV | | | | PRIVATE | | | |
|---|---|---|---|---|---|---|---|---|
| | AUC | F1 | Jaccard | wDDI | AUC | F1 | Jaccard | wDDI |
| LR | 0.632 ± 0.006 | 0.526 ± 0.004 | 0.401 ± 0.008 | 0.055 ± 0.0009 | 0.639 ± 0.007 | 0.561± 0.002 | 0.408 ± 0.004 | 0.146 ± 0.0004 |
| RF | 0.635 ± 0.004 | 0.531 ± 0.003 | 0.406 ± 0.004 | 0.053 ± 0.0005 | 0.642 ± 0.003 | 0.565 ± 0.005 | 0.411 ± 0.004 | 0.140 ± 0.0003 |
| BART | 0.637 ± 0.002 | 0.540 ± 0.002 | 0.408 ± 0.005 | 0.056 ± 0.0006 | 0.646 ± 0.004 | 0.571 ± 0.004 | 0.414 ± 0.002 | 0.142 ± 0.0005 |
| LEAP | 0.625 ± 0.003 | 0.514 ± 0.005 | 0.382 ± 0.006 | 0.047 ± 0.0003 | 0.633 ± 0.005 | 0.558 ± 0.007 | 0.405 ± 0.003 | 0.117 ± 0.0001 |
| DMNC | 0.633 ± 0.006 | 0.528 ± 0.002 | 0.397 ± 0.005 | 0.056 ± 0.0002 | 0.640 ± 0.007 | 0.566 ± 0.002 | 0.413 ± 0.004 | 0.158 ± 0.0005 |
| GameNet | 0.641 ± 0.003 | 0.543 ± 0.005 | 0.415 ± 0.003 | 0.053 ± 0.0005 | 0.647 ± 0.001 | 0.575 ± 0.004 | 0.421 ± 0.006 | 0.131 ± 0.0003 |
| CompNet | 0.638 ± 0.002 | 0.546 ± 0.006 | 0.413 ± 0.008 | 0.050 ± 0.0002 | 0.645 ± 0.005 | 0.582 ± 0.001 | 0.424 ± 0.006 | 0.134 ± 0.0005 |
| PREMIER | 0.655 ± 0.001 | 0.565 ± 0.002 | 0.421 ± 0.005 | 0.047 ± 0.0001 | 0.669 ± 0.001 | 0.607 ± 0.003 | 0.443 ± 0.004 | 0.122 ± 0.0001 |
| MICRON | 0.658 ± 0.003 | 0.567 ± 0.006 | 0.418 ± 0.002 | 0.047 ± 0.0006 | 0.662 ± 0.002 | 0.597 ± 0.006 | 0.438 ± 0.002 | 0.120 ± 0.0001 |
| SafeDrug | 0.652 ± 0.004 | 0.564 ± 0.002 | 0.415 ± 0.006 | 0.045 ± 0.0001 | 0.671 ± 0.003 | 0.608 ± 0.006 | 0.441 ± 0.008 | 0.113 ± 0.0002 |
| COGNet | 0.661 ± 0.002 | 0.573 ± 0.001 | 0.426 ± 0.001 | 0.054 ± 0.0006 | 0.682 ± 0.005 | 0.616 ± 0.002 | 0.453 ± 0.005 | 0.129 ± 0.0004 |
| MoleRec | 0.670 ± 0.004 | 0.579 ± 0.003 | 0.431 ± 0.002 | 0.044 ± 0.0001 | 0.688 ± 0.002 | 0.623 ± 0.006 | 0.459 ± 0.001 | 0.112 ± 0.0001 |
| **REFINE** | **0.708 ± 0.001\*** | **0.635 ± 0.003\*** | **0.462 ± 0.003\*** | **0.043 ± 0.0001\*** | **0.729 ± 0.004\*** | **0.656 ± 0.003\*** | **0.506 ± 0.002\*** | **0.111 ± 0.0002\*** |

\* indicates that the result is statistically significant when compared to the second best with p-value $< 0.05$.

Similarly, we vary the number of layers as well as the number of heads per layer of transformer encoders and report the performance of REFINE in Figure 2(b) and (c) respectively. We observe that having 4 layers with 2 attention head each performs best for both the datasets.

Next, we study the impact of varying the number of attention heads in the first layer of the two layer GATv2 used to model frequency based drug co-occurrence and severity based drug interaction information. We observe in Figure 2(d) that 4 attention heads lead to best performance on MIMIC-IV dataset while 2 attention heads provides optimal performance on the PRIVATE dataset.

Figure 3 shows the impact of loss weights $\gamma_1$ and $\gamma_2$ on the AUC, wDDI scores of REFINE for both MIMIC-IV and PRIVATE dataset. We observe that as $\gamma_1$ and $\gamma_2$ increase the AUC improves while the wDDI worsens for both datasets. For MIMIC-IV, $\gamma_1 = 0.75$ and $\gamma_2 = 0.07$ provides the desirable trade-off between AUC and wDDI. Similarly, $\gamma_1 = 0.80$ and $\gamma_2 = 0.05$ provides desirable trade-off for the PRIVATE dataset.

## D.2 Comparative Experiments

Table 1 shows the complete set of results (including Jaccard similarity) for comparative study with the standard deviations presented after ± in the tables.

We also analyze the impact of the number of visits on the performance of REFINE and two strong baselines COGNet and SafeDrug as shown in Figure 4. For this analysis, we choose a cohort of patients with 4 visits and 6 visits for MIMIC-IV and PRIVATE respectively. The number of visits is chosen based on the average number of visits per patient in each dataset. It is interesting to note that the gap in performance (especially in terms of AUC) between REFINE and the baselines increases with the number of visits. This shows that the trend information learned by REFINE helps provide a performance boost. Additionally, we see that REFINE outperforms COGNet and SafeDrug in terms of AUC and wDDI even when few visit instances are available. This shows that effective modeling of patient information along with the consideration of trade-off between benefits and potential interaction of drug pairs is important for recommending personalized, accurate and safe drug combinations.

## D.3 Ablation Study

Here, we perform ablation study to understand the impact of different knowledge sources on the performance of REFINE. To this end we implement the following variants of REFINE:

- **REFINE w/o $m_i$.** Here we use all the information sources except the medication dosages and their trends.

- **REFINE w/o $l_i$.** Here we use all the information sources except the lab test responses and their trends.

- **REFINE w/o med trend.** In this variant we only omit the trend information (variance and slope) obtained from the medication dosages.

- **REFINE w/o lab trend.** Here we only omit the trend information obtained from the lab test responses.

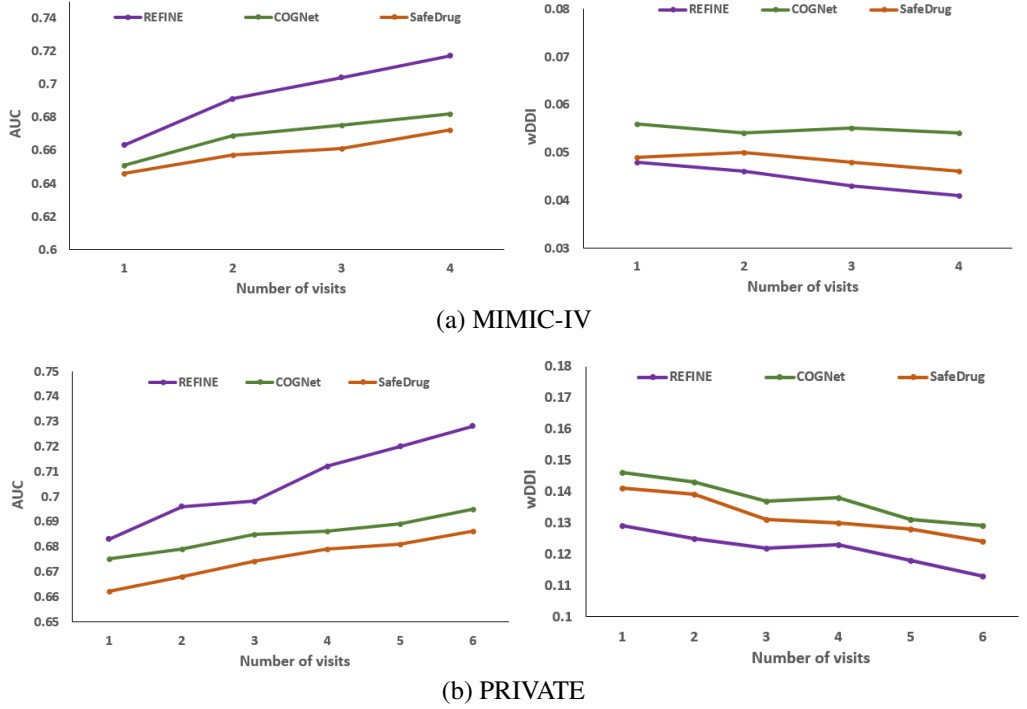

(a) MIMIC-IV

(b) PRIVATE

Figure 4: Effect of number of visits on performance of various methods.

Table 2: Results for ablation study.

| Methods | MIMIC-IV | | | | PRIVATE | | | |
|---|---|---|---|---|---|---|---|---|
| | AUC | F1 | Jaccard | wDDI | AUC | F1 | Jaccard | wDDI |
| w/o $m_i$ | $0.693 \pm 0.002$ | $0.610 \pm 0.002$ | $0.457 \pm 0.001$ | $0.0443 \pm 0.0002$ | $0.712 \pm 0.001$ | $0.640 \pm 0.003$ | $0.484 \pm 0.001$ | $0.1121 \pm 0.0002$ |
| w/o $l_i$ | $0.690 \pm 0.003$ | $0.605 \pm 0.002$ | $0.452 \pm 0.002$ | $0.0438 \pm 0.0003$ | $0.707 \pm 0.002$ | $0.635 \pm 0.001$ | $0.481 \pm 0.003$ | $0.1116 \pm 0.0001$ |
| w/o med trend | $0.701 \pm 0.001$ | $0.618 \pm 0.003$ | $0.460 \pm 0.002$ | $0.0440 \pm 0.0001$ | $0.718 \pm 0.003$ | $0.645 \pm 0.004$ | $0.489 \pm 0.002$ | $0.1118 \pm 0.0002$ |
| w/o lab trend | $0.695 \pm 0.002$ | $0.613 \pm 0.001$ | $0.456 \pm 0.003$ | $0.0444 \pm 0.0003$ | $0.714 \pm 0.004$ | $0.643 \pm 0.001$ | $0.487 \pm 0.004$ | $0.1120 \pm 0.0005$ |
| w/o any trend | $0.682 \pm 0.002$ | $0.602 \pm 0.002$ | $0.445 \pm 0.003$ | $0.0446 \pm 0.0002$ | $0.703 \pm 0.002$ | $0.630 \pm 0.001$ | $0.474 \pm 0.002$ | $0.1123 \pm 0.0004$ |
| w/o $G^C, G^D$ | $0.703 \pm 0.003$ | $0.629 \pm 0.004$ | $0.459 \pm 0.001$ | $0.0451 \pm 0.0002$ | $0.724 \pm 0.002$ | $0.652 \pm 0.004$ | $0.499 \pm 0.002$ | $0.1126 \pm 0.0001$ |
| w/o $L_{bdi}$ | $0.697 \pm 0.003$ | $0.621 \pm 0.004$ | $0.453 \pm 0.002$ | $0.0464 \pm 0.0003$ | $0.716 \pm 0.002$ | $0.642 \pm 0.001$ | $0.493 \pm 0.003$ | $0.1153 \pm 0.0002$ |
| w/o severity | $0.694 \pm 0.002$ | $0.617 \pm 0.001$ | $0.450 \pm 0.004$ | $0.0461 \pm 0.0005$ | $0.713 \pm 0.004$ | $0.639 \pm 0.002$ | $0.490 \pm 0.001$ | $0.1148 \pm 0.0003$ |
| w/o transformer | $0.696 \pm 0.003$ | $0.625 \pm 0.002$ | $0.448 \pm 0.001$ | $0.0439 \pm 0.0003$ | $0.718 \pm 0.002$ | $0.642 \pm 0.001$ | $0.495 \pm 0.003$ | $0.1119 \pm 0.0002$ |
| REFINE | $\mathbf{0.708 \pm 0.001^*}$ | $\mathbf{0.635 \pm 0.003^*}$ | $\mathbf{0.462 \pm 0.003^*}$ | $\mathbf{0.0431 \pm 0.0001^*}$ | $\mathbf{0.729 \pm 0.004^*}$ | $\mathbf{0.656 \pm 0.003^*}$ | $\mathbf{0.506 \pm 0.002^*}$ | $\mathbf{0.1112 \pm 0.0002^*}$ |

\* indicates that the result is statistically significant when compared to the second best with p-value $< 0.05$.

- **REFINE w/o any trends.** This variant represents a model without any trend information from both medication dosage and lab test response.

- **REFINE w/o $G^C$, $G^D$.** Here we use all the information sources except the drug co-occurrence and drug interaction graphs.

- **REFINE w/o $L_{bdi}$.** In this variant we only remove the $L_{bdi}$ loss and replace it with the traditional DDI loss [3] during the training of the model.

- **REFINE w/o severity.** In this variant we only replace the non-zero weight $w_{ij}^D$ with a value of 1 in the graph $G^D$. In other words, we do not use the severity information of the drug interaction.

- **REFINE w/o transformer.** In this variant we replace the Transformer modules with RNN modules.

Table 2 shows the complete set of results for the ablation study with the standard deviations presented after $\pm$ in the table. In addition to the discussion on the variants of REFINE in the main paper (see Section 4.2), it is interesting to note that **REFINE w/o $L_{bdi}$** records a drop in performance for both wDDI and accuracy. This suggests that our proposed loss $L_{bdi}$ helps to boost both the accuracy and safety of the recommendations. **REFINE w/o transformer** demonstrates a slight decrease in performance compared to its Transformer-based counterpart. This suggests that the improvement achieved by our proposed solution goes beyond merely employing Transformer modules.

Table 3: Results for comparative analysis on MIMIC-III.

| Methods | MIMIC-III | | | |
| --- | --- | --- | --- | --- |
| | AUC | F1 | Jaccard | wDDI |
| GameNet | $0.664 \pm 0.002$ | $0.566 \pm 0.004$ | $0.437 \pm 0.002$ | $0.0480 \pm 0.0003$ |
| SafeDrug | $0.671 \pm 0.001$ | $0.579 \pm 0.003$ | $0.441 \pm 0.001$ | $0.0429 \pm 0.0001$ |
| COGNet | $0.683 \pm 0.003$ | $0.591 \pm 0.002$ | $0.452 \pm 0.004$ | $0.0501 \pm 0.0002$ |
| MoleRec | $0.692 \pm 0.002$ | $0.606 \pm 0.003$ | $0.459 \pm 0.001$ | $0.0421 \pm 0.0004$ |
| REFINE | $\mathbf{0.730 \pm 0.002}$* | $\mathbf{0.658 \pm 0.001}$* | $\mathbf{0.487 \pm 0.002}$* | $\mathbf{0.0393 \pm 0.0002}$* |

* indicates that the result is statistically significant when compared to the second best with p-value $< 0.05$.

## D.4 Experiments on MIMIC-III

We also evaluate the performance of REFINE on the MIMIC-III [1] dataset. Table 3 compares the results with state-of-the-art. We can see that REFINE achieves the best performance with statistically significant improvements.