# OpenReview forum: "REFINE: A Fine-Grained Medication Recommendation System Using Deep Learning and Personalized Drug Interaction Modeling"
_NeurIPS.cc/2023/Conference — NeurIPS 2023 poster_

### Official Review · Reviewer_RMGb · 2023-06-19

**Soundness:** 2 fair
**Presentation:** 1 poor
**Contribution:** 1 poor
**Rating:** 6
**Confidence:** 5

**Summary:**

In this paper, the authors present a deep learning framework that utilizes patients' electronic health records and drug information to recommend safe and accurate medication combinations based on their symptoms and examination results. The proposed method is evaluated on two real-world datasets, MIMIC-IV and PRIVATE, and is compared with existing baselines. The results show that the proposed framework outperforms the baselines.

**Strengths:**

The topic is pioneering and meaningful. Learning to recommend safe and accurate drug combinations is an important component of intelligent healthcare, and it makes a significant contribution to the development of human healthcare.

**Weaknesses:**

- The presentation of the paper is poor, and the equations cause confusion for the reader. Additionally, the figures presented in the article are of low quality and are not vector graphics.
- The paper lacks necessary indicators and adequate comparisons with baselines, which undermines the overall validity of the experiment and the claimed contributions.

**Questions:**

**If the authors can provide satisfactory answers to my questions, I would consider revising my score.**

- (Major) The proposed method lacks evaluation in terms of "Jaccard", which is a metric used in many of the baselines such as SafeDrug (https://arxiv.org/abs/2105.02711) and GameNet (https://ojs.aaai.org/index.php/AAAI/article/view/3905). Evaluating the distances between the ground truth medication combination and the predicted one is vital, as many combinations of drugs require their presence to exert a synergistic effect.
- (Major) I find the validation to be insufficient. There should be comparisons to more state-of-the-art baselines such as MoleRec (https://dl.acm.org/doi/abs/10.1145/3543507.3583872) and 4SDrug (https://dl.acm.org/doi/abs/10.1145/3534678.3539089).
- (Major) The author declares that the suggested system is more potent than the baselines. Multiple baselines, such as SafeDrug (https://arxiv.org/abs/2105.02711)  and GameNet (https://ojs.aaai.org/index.php/AAAI/article/view/3905), utilize Recurrent Neural Networks (RNNs) to extract features from Electronic Health Records (EHRs) for patient representation, whereas the proposed system employs transformers. However, no experiment validates that the performance boost is due to the proposed system and not only a mere replacement of RNNs with transformers. To ensure a just and unbiased comparison, the author should compare the RNN version of the proposed system to demonstrate that the performance improvement is genuinely caused by the proposed system instead of transformers.
- (Major) MIMIC-IV benchmark AUC seems much lower than typical methods would produce, is it some mistake?  Reference: https://arxiv.org/abs/2305.12788. To my best knowledge, there does not seem to be any issue of out-of-distribution with the dataset. Comparing the result in the article to the reference, the reported performance of the baselines is noticeably lower by approximately **ten percentage** points. Despite the possibility of differences in train-valid-test dataset split, this discrepancy is still unusual.
- (Major) The author claimed that the balanced DDI Loss is powerful, there are no experiments demonstrating that it outperforms the DDI loss used in SafeDrug (https://arxiv.org/abs/2105.02711). Based on my understanding, the **REFINE w/o severity** in the ablation study removes the severity information from both the loss and the graph neural networks

- What's the meaning of GATv2 in Equation (4)? If it's a graph neural network, how can a GNN takes only one features as input instead of a graph; If it's not, what's the relation between it and the GATv2. Still how can the same linear transformation be applied to two vectors with different dimensions as it's shown in Equation (4) and Equation (5)? And does the two layers of GAT share the same parameters? It's claimed in the article that $\delta_{jk}$ is calculated via the average of $\delta_{jk}^b$.
- The figures should be presented in vector graphics. The information in Figure (4) should be presented in form of table instead of figures.
- Why the calculation of patient representation is obtained via $q_T=h_T^d+h_T^r+h_{T-1}^s$ instead of $q_T=h_T^d+h_T^r+h_{T}^s$？

If the author can respond satisfactorily to my questions, I am willing to adjust my score.

**Limitations:**

The article is poorly written and the experiment is not sufficient to support the contribution claimed in the article.

---

> ### Author Rebuttal · Authors · 2023-08-09
>
> We would like to thank the reviewer for highlighting the strong points and shortcomings of our work. We have addressed the queries as follows (numbers are in order of query raised):
>
> 1) Evaluation in terms of Jaccard as well as three other metrics have already been provided in Table 1 under Appendix I.
>
> 2) MoleRec has been shown to outperform 4SDrug in their paper. The table below shows the results of our comparative experiment with MoleRec. We see that REFINE significantly outperforms MoleRec across all metrics for both datasets.
>
> | Method                                          |   |   | MIMIC-IV |  |
> | -------------------------------------------- | ------------------------ | -----------| -----------| -----------|
> |  | AUC |	F1|	Jaccard |	wDDI |
> | __MoleRec__	                                |   0.670 ± 0.004	                |      0.579 ± 0.003 | 0.431 ± 0.002 | 0.044 ± 0.0001 |
> | __REFINE__		|  0.708 ± 0.001	                |      0.635 ± 0.003 | 0.462 ± 0.003 | 0.043 ± 0.0001 |
>
>
> | Method                                          |   |   | PRIVATE|  |
> | -------------------------------------------- | ------------------------ | -----------| -----------| -----------|
> |  | AUC |	F1|	Jaccard |	wDDI |
> | __MoleRec__	                                |   0.688 ± 0.002	                |      0.623 ± 0.006 | 0.459 ± 0.001 | 0.112 ± 0.0001 |
> | __REFINE__		|  0.729 ± 0.004	                |      0.656 ± 0.003 | 0.506 ± 0.002 | 0.111 ± 0.0002 |
>
> 3) The table below compares that the performance of REFINE when implement using both RNN and Transformer architectures. Although the RNN-based REFINE demonstrates a slight decrease in performance compared to its Transformer-based counterpart, the improvement in both cases is still more significant than the results achieved by the state-of-the-art method, MoleRec. This suggests that the improvement achieved by our proposed solution goes beyond merely employing Transformer modules.
>
> | Method                                          |   |   | MIMIC-IV |  |
> | -------------------------------------------- | ------------------------ | -----------| -----------| -----------|
> |  | AUC |	F1|	Jaccard |	wDDI |
> | __MoleRec__	                                |   0.670 ± 0.004	                |      0.579 ± 0.003 | 0.431 ± 0.002 | 0.0442 ± 0.0001 |
> | __REFINE with RNN__		|  0.696 ± 0.003	                |      0.625 ± 0.002 | 0.448 ± 0.001 | 0.0439 ± 0.0003 |
> | __REFINE with Transformer__		|  0.708 ± 0.001	                |      0.635 ± 0.003 | 0.462 ± 0.003 | 0.0431 ± 0.0001 |
>
> | Method                                          |   |   | PRIVATE|  |
> | -------------------------------------------- | ------------------------ | -----------| -----------| -----------|
> |  | AUC |	F1|	Jaccard |	wDDI |
> | __MoleRec__	                                |   0.688 ± 0.002	                |      0.623 ± 0.006 | 0.459 ± 0.001 | 0.1123 ± 0.0001 |
> | __REFINE with RNN__		|  0.718 ± 0.002	                |      0.642 ± 0.001 | 0.495 ± 0.003 | 0.1119 ± 0.0002|
> | __REFINE with Transformer__		|  0.729 ± 0.004	                |      0.656 ± 0.003 | 0.506 ± 0.002 | 0.1112 ± 0.0002 |
>
> 4) The work in https://arxiv.org/abs/2305.12788 recommends medications at the class level, which is a coarse-grained approach. In contrast, our work recommends medications at the drug level, which is a more fine-grained approach. Hence, we cannot directly compare the AUC for the Precision-Recall curve (and other metrics) between our method/baselines and the cited article. Recommending specific drugs introduces a greater complexity compared to class-level recommendations, as it requires consideration of whether patients need single or multiple drugs from the same class.
>
>
> 5) An evaluation of our system's performance, utilizing the DDI loss found in SafeDrug (labeled as REFINE w/o L_bdi)), can be found in the full Ablation study (Table 2 under Appendix J). We noted that substituting our balanced DDI loss with the DDI loss used in SafeDrug led to a decline in performance. This was observed both in terms of accuracy and the degree of drug interaction within the recommended medication set.
>
> 6) In Equation (4), GATv2 in used to depict the application of feed-forward neural network layer after the LeakyReLU function, as well as the linear transformation layer after the concatenation process. To avoid confusion, we have revised Equation (4) as follows:
> $$
> δ_{jk}^b=  \frac{w_{jk}^D  ×exp⁡(a^{'}LeakyReLU(E^b∙[n_j ||n_k]))}{∑_{n_{i} ϵS_{j}}w_{ij}^D  × exp⁡(a^{'}LeakyReLU(E^b∙[n_j ||n_i])) }
> $$
> $a^{'}$ is the transpose of the weight vector of a single layer feed-forward neural network, $E^b=[E^{b^* } || E^{b^* }]$  as shown in https://arxiv.org/pdf/2105.14491.pdf where $E^{b^* }$  is the embedding matrix for the $b^{th}$ attention head. Equation (5) is also revised to:
> $$
> o_j^D=∥_{b=1}^z σ(∑_{n_k ϵS_j} δ_{jk}^b E^{b^* }∙ n_k )
> $$
> No, the layers do not share the same parameters. In the second layer, for calculating $ δ_{jk} $ we update $ w_{jk}^D  $ to average of $ δ_{jk}^b $ in equation (4). In other words, the edge weights for the second layer are updated with average (across heads) attention coefficients of the first layer.
>
> 7) Yes, we will update the figures as suggested in the camera-ready paper.
>
> 8) In this work, $h_i^d , h_i^r, h_i^s$ depict the representations of diagnosis, lab test response, medication dosage set obtained from the transformer at the $i^{th}$ visit. Since our goal is to recommend a medication set for visit T, we are limited to utilizing the medication information up to visit  T-1, that is, $h_{T-1}^s$.

---

> > ### Comment · Reviewer_RMGb · 2023-08-10
> > **Thanks for your response!**
> >
> > Thank you for your response. Your feedback has addressed most of my concerns, and I have decided to raise my score to 5. However, it seems that the button to provide editing review comments is not yet available. I will adjust the score accordingly once the option becomes accessible.
> >
> > Below are some suggestions regarding the paper's formatting:
> >
> > - The original Figures 1, 2, and 3 in the paper occupy significant space while conveying minimal information. I recommend that the author consolidate and redraw these figures to convey their content more effectively. Additionally, consider moving the ablation study section from the appendix to the main body of the article to enhance its content. Moreover, it would be beneficial if the author could reformat and compress the tables in the article to reduce whitespace. I hope the author can incorporate more meaningful experimental results from the appendix into the main text.
> >
> > - To ensure the completeness and persuasiveness of the article, I suggest that the author include the supplementary experiments conducted during the rebuttal process in the subsequent revision.
> >
> > In addition, considering that the baselines mentioned in the paper, such as GameNet and Safedrug, as well as the newly introduced MoleRec, conducted their experiments using the MIMIC-III dataset in their original papers. While it's possible that the MIMIC-IV dataset used in this paper might include MIMIC-III data, I am still looking forward to seeing relevant results. If the authors could supplement these experiments and demonstrate the effectiveness of the proposed method, I would consider raising my score further from the previously mentioned 5 points.

---

> > > ### Author Response · Authors · 2023-08-12
> > >
> > > We would like to thank the reviewer for the response and the suggestions. We have taken note of the paper's formatting suggestions and will incorporate them into the paper. MIMIC-III was collected for patients admitted to BIDMC between 2001 and 2012 while MIMIC-IV was collected for patients admitted to BIDMC between 2008  and 2019. Hence, we believe there might be a large overlap of patients between the datasets.
> > >
> > > Nonetheless, we are currently conducting our experiments on the MIMIC-III dataset and will share our findings soon.

---

> > > > ### Author Response · Authors · 2023-08-15
> > > > **Additional experiments on MIMIC-III**
> > > >
> > > > We provide the results on the MIMIC-III dataset in the table below. From the table we can see that REFINE outperforms the baselines by a significant margin for MIMIC-III (wDDI 0.0401) dataset as well.
> > > > | Methods|	| MIMIC-III |  |  |
> > > > | -------------| -------------------| ------------ | ------------------------ | -----------|
> > > > |	| AUC |	F1	| Jaccard	| wDDI |
> > > > |__GameNet__ |	0.664 ± 0.002 |	0.566 ± 0.004	| 0.437 ± 0.002 |	0.0480 ± 0.0003
> > > > |__SafeDrug__ |	0.671 ± 0.001 	| 0.579 ± 0.003 	| 0.441 ± 0.001 |	0.0429 ± 0.0001
> > > > | __COGNet__ |	0.683 ± 0.003  |	0.591 ± 0.002 |	0.452 ± 0.004 |	0.0501 ± 0.0002
> > > > | __MoleRec__ |	0.692 ± 0.002 | 	0.606 ± 0.003 |	0.459 ± 0.001 |	0.0421 ± 0.0004
> > > > | __REFINE__ | 	0.730 ± 0.002 |	0.658 ± 0.001 |	0.487 ± 0.002 |	0.0393 ± 0.0002
> > > >
> > > > Do let us know if you have any queries.

---

> > > > > ### Comment · Reviewer_RMGb · 2023-08-15
> > > > >
> > > > > Thank you for providing the experiments on the MIMIC-III dataset. I appreciate the authors addressing this aspect of the research. I hope that in the subsequent revisions, the authors will further integrate these experiments to highlight the model's performance.
> > > > >
> > > > > I have revised my score from 5 to 6.

---

> > ### Comment · Reviewer_RMGb · 2023-08-11
> >
> > Dear authors, I have raised my score to 5 points. Best of luck to you all.

---

### Official Review · Reviewer_Eh21 · 2023-07-02

**Soundness:** 3 good
**Presentation:** 3 good
**Contribution:** 2 fair
**Rating:** 5
**Confidence:** 5

**Summary:**

This paper proposes a new medication recommendation model that learns patient representations from diagnoses, lab tests, and medications. The new aspect of the paper is that it considers different severities of drug-drug interactions and can prevent overly conservative drug prescriptions.

**Strengths:**

1. The biggest strength of this paper is that it utilizes the DDInter databases into the drug recommendation tasks. As mentioned in the paper, the DDInter database has broken down the drug drug interactions into different severity levels. Previous DDI-aware drug recommendation models, such as SafeDrug, only considers the binary DDI existence and can provide overly conservative prescriptions. By considering the severity level, the proposed model can improve the performance by including some drug pairs that are beneficial and have low co-prescription severity.

2. The paper is well written and the experiments are extensive. Figures in the paper are very informative.

**Weaknesses:**

1. The contribution of the paper is below the NeurIPS acceptance bar.
2. Some statements are not well supported or overclaimed.
    - In the conclusion, the paper states that the proposed model can offer personalized medication suggestions. It is not discussed in the main paper.
    - In the conclusion, the paper states that REFINE is the only system that factors in the medication, lab test trends. This is not true. Many other models, such as GAMENet, also model the progression of medications, diagnoses, etc.
3. Some details in the paper is missing.
    - what is $\mu_i^t$ in equation (2)
    - how does equation (8) work? $q_T$ is a vector and $Z\times \lambda$ is a matrix.
    - should equation (9) be $\beta\times xxx + (1-\beta)\times xxx$. Use the plus sign?
    - what is the hinge loss?
    - why choose 0.5 as the medication recommendation threshold? It seems that each binary classification is highly imbalanced, and thus the learned logits are more skewed towards 0. Maybe a threshold of 0.3 would be a better choice.
4. Two datasets have a very small medication space (one has 735 medications, and another has 50). Are they individual drugs? As far as the reviewer knows, MIMIC-IV can have more than 10,000+ drug IDs. If the paper claims on individual drug recommendations (as stated in the intro), then this part needs explanation.


**Questions:**

see the above.

---

> ### Author Rebuttal · Authors · 2023-08-09
>
> We would like to thank the reviewer for the thoughtful reviews. We have addressed some of the queries and have summarized as follows:
>
> 1) The contribution of the paper is below the NeurIPS acceptance bar.
>
> We would like to emphasize that our paper introduces an innovative fine-grained medication recommendation system, addressing actual challenges and needs.
> The main highlights of our work include:
>
> Learning patient condition progression representation:  Beyond the sequential dependencies modelled by existing techniques, we also analyze and model fluctuations in medication dosage titrations and lab test responses over the time between visits. This offers a robust representation of a patient's evolving condition.
>
> Incorporating drug interaction severity: Unlike current methods, we embed information regarding drug severity into our recommendations. Furthermore, we formulate a balanced DDI loss, accounting for both the level of drug severity and the frequency of co-occurrence within the EHR. This leads to recommendations more closely aligned with clinical practices.
>
> 2) Some statements are not well supported or overclaimed.
>
>   2a) In the conclusion, the paper states that the proposed model can offer personalized medication suggestions. It is not discussed in the main paper.
>
> Our proposed system is designed to recommend medications by taking into account a patient's current conditions along with their medical history. By specifically modeling trends in medication dosages and lab test responses, our system is capable of personalizing medication recommendations tailored to individual needs. We recognize the oversight and will include this discussion in the main paper.
>
>   2b) In the conclusion, the paper states that REFINE is the only system that factors in the medication, lab test trends. This is not true. Many other models, such as GAMENet, also model the progression of medications, diagnoses, etc.
>
> While models like GAMENet do model the sequential dependency of medications or diagnoses over visits, our approach goes a step further. We specifically factor in the trends of medication dosages and lab test responses over time, thus capturing a more detailed progression. This distinction sets our model apart in its approach to analyzing these medical trends.
>
> 3) Some details in the paper is missing.
>
>   3a) what is $μ_i^t$ in equation (2)
>
> It should be $t_i^{ ̅}$ which depicts the average $t_j$ for j ∈ [1, i]. We have corrected the equation.
>
>  3b) How does equation (8) work? $q_T$ is a vector and Z × λ is a matrix.
>
> Suppose $d$ is the embedding dimension and $N_m$ is the total number of drugs, then $q_T$ is a vector of dimension $d×1$, $Z$ is a matrix of dimension $d× N_m$ and $λ$ is a vector of dimension $N_m×1$. Therefore $Z∙λ$ will yield a vector of dimension $d×1$ .
>
> 3c) should equation (9) be $β ×xxx+(1-β)×xxx$. Use the plus sign?
>
> No, plus sign should not be used in this context. Our goal is to avoid over-conservative recommendation by allowing drug pairs that have frequent co-occurrences though they may have higher severity interaction. Hence, we use the minus sign to favour drug pairs with high co-occurrences.
>
> 3d) what is the hinge loss?
>
> We have defined hinge loss in Appendix C.
>
> 3e) why choose 0.5 as the medication recommendation threshold? It seems that each binary classification is highly imbalanced, and thus the learned logits are more skewed towards 0. Maybe a threshold of 0.3 would be a better choice.
>
> We have experimented with different threshold values. The use of hinge loss helps maintain the threshold at 0.5 (see Appendix C).
>
>
> 4) Two datasets have a very small medication space (one has 735 medications, and another has 50). Are they individual drugs? As far as the reviewer knows, MIMIC-IV can have more than 10,000+ drug IDs. If the paper claims on individual drug recommendations (as stated in the intro), then this part needs explanation.
>
> The small medication space referenced in our datasets refers to individual generic drug names, not specific drug IDs. We translated the NDC codes found in the MIMIC-IV dataset into medication generic names using the mapping tables outlined in Appendix F. Although there are 5733 unique NDC codes in MIMIC-IV dataset (in the file prescriptions.csv), not all these codes have a one-to-one mapping with a medication generic name. This is because the same medication with different dosages/manufacturers will have different NDC codes, e.g., Metformin can have multiple NDC codes based on the dosage strength or the manufacture. Further, we use the medications prescribed within the first 24 hours of each patient's visit. This timeframe is considered critical in patient care and has also been adopted in other baseline studies, such as GameNet. We believe that these factors, coupled with the other pre-processing steps detailed in Appendix F, reduced the size of the medication set.

---

> > ### Author Response · Authors · 2023-08-21
> >
> > We would like to thank the reviewer for the detailed comments and suggestions and hope we have addressed all your concerns. If you have any other queries or concerns we would be happy to clarify/discuss them.

---

### Official Review · Reviewer_ebob · 2023-07-05

**Soundness:** 3 good
**Presentation:** 4 excellent
**Contribution:** 3 good
**Rating:** 6
**Confidence:** 4

**Summary:**

This paper presents a medication recommendation system with 2 components: a GAT based medication interaction and a ViT-based patient embedding. The results on MIMIC and PRIVATE shows its superior performance.

**Strengths:**

1: I think this paper is well-presented. The structure is very clear and the writing is polished.
2: The results are generalizable -- consistent performance can be observed from two datasets.
3: The experiments are comprehensive. The qualitative study is also interesting.

**Weaknesses:**

1: Reproducibility: The codes/notebooks are not shared either through supplementary materials or anonymous github. The paper does not provide much detail about the implementation.
2: The choice of baselines: For EHR dataset, it is well-known that tree-based methods (like Gradient Boosting Tree, Random Forest, etc) perform the best. There are also some causal inference tree models like Causal Forest and BART. It is strange to me that the authors include LR but neglect all tree models.
3: Give that the proposed work is a concatenation of ViT and GAT, the level of novelty is questionable.

**Questions:**

1: I really think you need to share the Jupyter Notebook (or codes) in whatever way to convince readers that the work is reproducible. It is very hard to implement simply based on the model description.
2: How does the small improvement in AUC translate to clinical impact? You need to sell it to the readers.
3: I encourage the authors to add some visualization figures. For example, how does the patient embedding look like, or how are the drug interations distributed?

---

> ### Author Rebuttal · Authors · 2023-08-09
>
> We would like to thank the reviewer for highlighting the strong points and shortcomings of our work. We have addressed some of the queries and have summarized them as follows:
>
> 1) Given that the proposed work is a concatenation of ViT and GAT, the level of novelty is questionable.
>
> We would like to emphasize that our paper introduces an innovative fine-grained medication recommendation system, addressing actual challenges and needs.
> The main highlights of our work include:
>
> Learning patient condition progression representation:  Beyond the sequential dependencies modelled by existing techniques, we also analyze and model fluctuations in medication dosage titrations and lab test responses over the time between visits. This offers a robust representation of a patient's evolving condition.
>
> Incorporating drug interaction severity: Unlike current methods, we embed information regarding drug severity into our recommendations. Furthermore, we formulate a balanced DDI loss, accounting for both the level of drug severity and the frequency of co-occurrence within the EHR. This leads to recommendations more closely aligned with clinical practices.
>
>
>
> 2)	I really think you need to share the Jupyter Notebook (or codes) in whatever way to convince readers that the work is reproducible. It is very hard to implement simply based on the model description.
>
> We will publish our code on Github upon acceptance of the paper. Currently, we have provided the data pre-processing and the training/optimization details in Appendix F and H respectively.
>
> 3)	The choice of baselines: For EHR dataset, it is well-known that tree-based methods (like Gradient Boosting Tree, Random Forest, etc) perform the best. There are also some causal inference tree models like Causal Forest and BART. It is strange to me that the authors include LR but neglect all tree models.
>
> We conducted additional experiments to compare the performance of tree-based methods (Random Forest) with REFINE. The table below shows substantial improvement achieved by REFINE across all metrics in both datasets.
>
>
> | Method                                          |   |   | MIMIC-IV |  |
> | -------------------------------------------- | ------------------------ | -----------| -----------| -----------|
> |  | AUC |	F1|	Jaccard |	wDDI |
> | __Random Forest__	                                |   0.635 ± 0.004	                |      0.531 ± 0.003 | 0.406 ± 0.004 | 0.053 ± 0.0005 |
> | __REFINE__		|  0.708 ± 0.001	                |      0.635 ± 0.003 | 0.462 ± 0.003 | 0.043 ± 0.0001 |
>
>
> | Method                                          |   |   | PRIVATE|  |
> | -------------------------------------------- | ------------------------ | -----------| -----------| -----------|
> |  | AUC |	F1|	Jaccard |	wDDI |
> | __Random Forest__	                                |   0.642 ± 0.003	                |      0.565 ± 0.005 | 0.411 ± 0.004 | 0.140 ± 0.0003 |
> | __REFINE__		|  0.729 ± 0.004	                |      0.656 ± 0.003 | 0.506 ± 0.002 | 0.111 ± 0.0002 |
>
>
> 4)	How does the small improvement in AUC translate to clinical impact? You need to sell it to the readers.
>
> We assess the performance of our model through the AUC for the precision-recall curve, where a higher value signifies fewer missed medications and a reduced number of extra recommended medications. In a perfect system, the AUC would reach 1, implying no missed or extra medications. In the case of MIMIC-IV with 10,218 patients and  40,872 visits (in the test set), an improvement of 0.047 in the AUC over COGNet can correspond to a substantial increase in correct medication recommendations across numerous patients. Such improvements not only translate into more accurate and personalized treatment but also contribute to overall enhancements in healthcare quality.
>
> 5)	I encourage the authors to add some visualization figures. For example, how does the patient embedding look like, or how are the drug interactions distributed?
>
> The visualization of the distribution of the number of visits in both datasets is provided in Figure 4 in the Appendix. We have included the figures showing the distribution of drug interactions in both datasets in the pdf attached to the global response.

---

> > ### Comment · Reviewer_ebob · 2023-08-16
> >
> > Thank you for the response. I would keep my rating for now. If the authors can include causal forest (not random forest) and bart and share the notebooks demonstrating the results, I would then lift up my score.

---

> > > ### Author Response · Authors · 2023-08-21
> > >
> > > Here are the results for BART:
> > > | Method                                          |   | MIMIC-IV	|   |
> > > | -------------------------------------------- | ------------      |------------------------ | -----------|
> > > | | F1| Jaccard | wDDI|
> > > | __BART__	                                |   0.540 ± 0.002 |	0.411 ± 0.003 |	0.056 ± 0.0006 |
> > > | __REFINE__		                |     0.635 ± 0.003 | 	0.462 ± 0.003 |	0.043 ± 0.0001 |
> > >
> > > | Method                                          |   | PRIVATE	|   |
> > > | -------------------------------------------- | ------------      |------------------------ | -----------|
> > > | | F1| Jaccard | wDDI|
> > > | __BART__	                                |   0.571 ± 0.004 |	0.417 ± 0.002 | 	0.142 ± 0.0005 |
> > > | __REFINE__		                |     0.656 ± 0.003 | 	0.506 ± 0.002 | 	0.111 ± 0.0002 |
> > >
> > > We observe that REFINE outperforms BART by a large margin. The adaptation of causal forest, which is generally used in treatment effect estimation, for our task of medication recommendation requires more time. We will include the results in the camera-ready copy. Meanwhile, the codes will be shared with the reviewers on an anonymous github by today.
> > >
> > > We appreciate the reviewer's detailed comments, and hope we have addressed all your concerns.

---

> > > > ### Author Response · Authors · 2023-08-21
> > > > **Code Repository**
> > > >
> > > > Dear Reviewer,
> > > >
> > > > We have shared the code repository for your consideration.
> > > > https://anonymous.4open.science/r/REFINE_codes-0206/
> > > >
> > > > This contains code for our work along with BART.
> > > >
> > > > Hope this addresses your concern. Thank you.

---

> > > > > ### Comment · Reviewer_ebob · 2023-08-21
> > > > >
> > > > > Thank you. I will lift up my score now. Good luck!

---

### Official Review · Reviewer_nz6X · 2023-07-06

**Soundness:** 2 fair
**Presentation:** 3 good
**Contribution:** 2 fair
**Rating:** 6
**Confidence:** 3

**Summary:**

This manuscript proposes a new system for personalized medication recommendation.  The method relies on the history of patient visits: diagnoses, lab responses and medication dosages across time. For obtaining safe recommendations, it uses severity level information of drug interactions. An important aspect of the model is the extraction of trend information, measuring how much medication dosages and lab responses change across patient visits. The approach relies on deep learning for obtaining drugs’ and patients’ representations that are then combined. More specifically, it uses transformers for learning a patient’s representation, and graph attention networks for learning drug interaction representations.

One of the main contributions of the paper is modelling drug interaction severity, and providing fine-grained personalized recommendations, that is, personalized recommendations at the drug level. It combines existing approaches and adapts graph attention networks for weighted graphs containing drug interaction severity information. In addition, the authors designed a balanced drug interaction loss function that weighs the benefits of a pair of drugs against the severity of any potential drug interaction.

**Strengths:**

- The problem is difficult and relevant

- The proposed approach is sound

- The proposed method has a better performance than state-of-the art methods

- The authors claim that REFINE is the first personalized medication recommendation systems to make predictions at the drug level

- They show the importance of including the drug interaction severity information

- The authors adapted attention weights for their problem

- They designed a balanced drug interaction loss function that weighs the benefits of a pair of drugs against the severity of any potential drug interaction

- The writing is clear.

**Weaknesses:**

**Major**

- The paper did not provide the code used in the experiments. Notice, however, that the authors mentioned in the neurips form that they would make it available “soon”.

- There are some aspects of the model that I did not fully understand. There are details missing about the dataset and how the information is represented. I will explain my concerns in the Questions section.

- The paper relies on a private dataset containing patient data. I think it is important to indicate whether all ethics concerns were addressed.

**Minor**

- I suggest some improvements in the mathematical notation. In 3.3, node and node’s representation of a drug $i$ are denoted in the same way ($n_i$). Could the authors use different notation and clarify whether $n_i$ is a node embedding in Eq 4?

- The authors use set notation {$1, N_m$} when they refer to intervals. For example, in lines 170, and 179, do the authors mean $[1, N_m]$ or {$1, 2, …, N_m$} instead of {$1, N_m$}?

- Reference 4 refers to an arXiv version of the paper. The paper was published in ICLR 2022. I recommend also checking if references 22 and 23 were published

- The venue where reference 5 was published is missing

**Questions:**

1. Previous approaches argue that they do class-level predictions for making the data compatible with MIMIC. Reference 16 uses the same version of the database as in this paper (MIMIC-IV). I am not used to this dataset. Could the authors explain why did they manage to make drug-level predictions using MIMIC database whereas the other methods did not? Are the NDC codes available for most of the drugs?

2. For each patient visit, the model receives as input a multi-hot encoding vector of diagnoses, lab test response vector and medication dosage vector. Could the authors answer the questions below?

    - (2.a) I am assuming that patients may have different types of exams and take different medications. I am also assuming that all patients share the same transformers. How is the input represented so that it is comparable across different patients? Could the authors clarify this in the paper?

    - (2.b) Also, could the authors clarify what they mean by “for the lab test response, only the first entry of  a lab test for a given visit is considered” in Section F Data Pre-Processing and Dataset Distribution, in Appendix. Is only the first exam result considered for a visit? In the main paper, the authors mention that each visit has a list o lab responses, so I was confused when I read the appendix. What is each entry of the a lab test response? Do the entries have a particular order? I recommend adding more details about how lab responses are pre-processed and represented in the model.

3. The input of GATv2 corresponds to initial node embeddings. How are they initialized? If they are random, what are the range of values and distribution used?

4. One of the datasets used by the model is private. I noticed in Table 1 that it is very different from the public dataset. Could the authors explain why? Are the public and private datasets focused on different types of patients/diseases? If the private dataset refers to a particular group of patient or type of disease, I would recommend the authors to include this information in the paper.

5. In the comparative analysis, Section 4.1, the authors mention that their method have a lower wDDI than in the original datasets (MIMIC and PRIVATE). I was wondering if it is necessarily a good thing. Could it mean that the predictions are too conservative and avoiding drug combinations that could still be important for the patient?

**Limitations:**

The authors addressed some limitations of their work.

Another aspect that could be addressed is how safe it is to apply the proposed system in a real world scenario. Would a doctor trust the recommendations made by the system? What would be the main impacts of using this system? In which scenarios do the authors think it would be more useful?

---

> ### Author Rebuttal · Authors · 2023-08-09
>
>
> We would like to thank the reviewer for the thorough analysis and comments on our work. We have addressed the queries as follows:
>
> 1) Existing works have focused on coarse-grained medication recommendations at the class level since making drug-level predictions introduces more complexity. In the MIMIC datasets, the medication names are noisy. However, NDC codes are provided for almost all drugs. This allows us to derive the corresponding medication names from the NDC codes using RxNorm mappings.
>
> 2.a) Each patient is represented by fixed length vectors for diagnosis, lab test responses, medication dosages based on the respective vocabulary size in the dataset. Suppose there are two lab tests [i,j]  and three medications [a,b,c] in the dataset, then the lab test vector has 2 entries while the medication vector has 3 entries and the values in these vectors depict the respective lab test response or medication dosage.
> For example, suppose a patient has undergone lab test i and is prescribed two medications (a, c), his lab test vector is  [0.8, 0] and medication vector is [0.3, 0, 0.4]. For another patient with lab test j and medication b, we will have a lab test vector of [0, 0.2] and a medication vector of [0, 0.6, 0].
>
> 2.b) In the dataset, some lab tests are repeated more than once in a visit, and we consider the initial lab test response as indicative of the effect of the treatment regime of the previous visit. Each position in the lab test response vector corresponds to a specific lab test, and each entry in the vector captures the results of the initial lab test during the visit.
>
> 3) The embedding matrices are initialized using xavier uniform initialization with a gain of 1.41 as described in Understanding the difficulty of training deep feedforward neural networks - Glorot, X. & Bengio, Y. (2010).  $n_i$ is a one-hot vector that depicts the presence of the drug i.
>
> 4) MIMIC dataset consist of data related to inpatient care, particularly ICU patients, whereas the Private dataset contains outpatient information with chronic diseases such as diabetes, and hypertension. We will add this information to the paper.
>
> 5) Table 4 in the main paper shows that the average number of medications missed by REFINE is the lowest, indicating that the prediction by REFINE is not too conservative and has not avoided drug combinations that are important for the patient.
>
> Additional Queries:
>
> 6) The paper relies on a private dataset containing patient data. I think it is important to indicate whether all ethics concerns were addressed.
>
> We have obtained IRB approval to use the dataset for this research, and no ethics regulations were violated.
>
> 7) I suggest some improvements in the mathematical notation. In 3.3, node and node’s representation of a drug i are denoted in the same way ($n_i$). Could the authors use different notation and clarify whether ni is a node embedding in Eq 4?
>
>  $n_i$ depicts a drug i representation and not the node embedding. Here, $n_i$ is a one hot vector which depicts the presence of a drug i. We will clarify this in the paper.
>
> 8) The authors use set notation {1, $N_m$} when they refer to intervals. For example, in lines 170, and 179, do the authors mean [1, $N_m$] or {1, 2, … , $N_m$} instead of {1, $N_m$}?
>
> We mean {1, 2, … , $N_m$} and have updated the paper accordingly.
>
> 9) Reference 4 refers to an arXiv version of the paper. The paper was published in ICLR 2022. I recommend also checking if      references 22 and 23 were published
>
> We have updated the paper accordingly.
>
> 10) The venue where reference 5 was published is missing.
>
> We have noted this and have updated the paper accordingly.
>
> 11) Another aspect that could be addressed is how safe it is to apply the proposed system in a real world scenario. Would a doctor trust the recommendations made by the system? What would be the main impacts of using this system? In which scenarios do the authors think it would be more useful?
>
> We believe that our proposed system can provide safe recommendations in the real-world scenario due to the use of a balanced drug interaction loss which helps ensures the safety and accuracy of the recommendations as shown in the experiments and case studies from real-world datasets.
>
> During our collaboration and interactions with the clinicians, we found that they were keen to use medication recommender systems provided the recommendations were at the drug level. This is in fact what motivated us to come up with a fine-grained medication recommendation system. Our proposed system provides contribution scores to identify factors influencing the recommendations as described in Appendix B and as demonstrated in the case studies (Section 5), which establishes trust among clinicians.
>
> Considering a patient’s history along with keeping track of interacting drug pairs to prescribe a tailor-made treatment regime is a challenging task for clinicians. Hence, our proposed system provides decision support to clinicians to make accurate and safe recommendations. Moreover, such a system would help reduce the high workload on clinicians by helping them make faster and more precise decisions.
>
> We believe that our proposed system will be more useful in providing personalized recommendations to help clinicians to design treatment regimens to better manage chronic diseases (diabetes, hypertension, etc.).

---

> > ### Comment · Reviewer_nz6X · 2023-08-15
> > **Thank you for addressing most of my questions!**
> >
> > Dear authors,
> >
> > Thank you for your response!
> >
> > Most of my main concerns were properly addressed.  However, I did not raise my score, because I could not check the code. This will be crucial for my review.
> >
> > I was wondering if the authors could upload their code during this discussion period.
> >
> > Below I write some comments and minor suggestions.
> >
> > 1. I thank the authors for clarifying some details about the method and the dataset. I believe that  the explanation that some lab tests can be repeated on the same visit, and about how the embedding matrices are initialized are very helpful. I was wondering if this could be added to the paper.
> >
> > 2. Regarding the discussion on wDDi metric, I agree that REFINE is less conservative than competitors. However, when the authors point out that the value of wDDi obtained by REFINE is lower than in the original dataset, I was wondering if it is necessarily a good thing. For example, we do not know if doctors chose riskier combinations than REFINE by mistake or if these combinations were essential for the patient's treatment. Nevertheless, I understand that the main conclusion of the analysis of the wDDI metric is that REFINE tends to make safe predictions, which is well supported by the paper.
> >
> > 3. Regarding the ethical concerns, as I do not have expertise in this, I will ask the area chairs if they were addressed accordingly.

---

> > > ### Author Response · Authors · 2023-08-15
> > >
> > > We are glad that we have been able to address most of your key concerns.
> > >
> > > We understand the importance of code accessibility for the validation and replication of our results by the community and are committed to ensure the integrity of our work with plans to release the code once our paper is accepted.
> > >
> > > Meanwhile, if there are specific parts of the experiments/methods that you believe need more clarity, please let us know. We are more than willing to provide the details and, if needed, share that specific portion of the code.

---

> > > > ### Comment · Reviewer_nz6X · 2023-08-15
> > > > **Raising my score to 5**
> > > >
> > > > Thank you for considering sharing parts of the code during the review period.
> > > >
> > > > The main reason why I would like to see the code is for evaluating the reproducibility of the work.
> > > >
> > > > I will raise my score to 5 as my other concerns were addressed.

---

> > > > > ### Author Response · Authors · 2023-08-21
> > > > > **Code repository**
> > > > >
> > > > > Dear Reviewer,
> > > > >
> > > > > We understand your concern and have shared the code repository for your perusal.
> > > > > https://anonymous.4open.science/r/REFINE_codes-0206/
> > > > >
> > > > > Hope this addresses your concern.

---

> > > > > > ### Comment · Reviewer_nz6X · 2023-08-22
> > > > > > **Thank you for sharing the code!**
> > > > > >
> > > > > > Thank you for sharing the code. I will increase my score to 6. Good luck!

---

### Author Rebuttal · Authors · 2023-08-09

Response to ethical concerns.

We take data privacy and security very seriously, and we would like to reassure the reviewer that all necessary precautions were taken to ensure that no ethics regulations were violated in the use of the private dataset for this research. The data has been appropriately anonymized and processed in compliance with all relevant ethical guidelines and legal requirements.

We have attached a pdf with figures showing the distribution of drug interactions in both datasets.

---

### Author Response · Authors · 2023-08-21

We would like to thank all the reviewers for their suggestions. Below we provide the code repository:
https://anonymous.4open.science/r/REFINE_codes-0206/

Hope it helps.

---

### Decision · Program_Chairs · 2023-09-21

**Decision:**

Accept (poster)

**Comment:**

The paper presents a deep learning framework focused on medication recommendations, a subject of high importance in healthcare. While the work is acknowledged for its potential impact, reviewers have varied opinions on its overall quality. Strengths of the paper include its clear structure, polished writing, and the incorporation of severity levels of drug-drug interactions from the DDInter database, which improves its applicability across multiple datasets.

However, the paper faces substantial criticisms across different aspects. There are concerns about the clarity and quality of equations and figures and insufficient validation against state-of-the-art baselines. The paper is also criticized for not providing enough evidence to support its claim of superiority over existing methods, particularly regarding whether its advantages stem from the proposed framework or merely from the technology used, such as transformers over RNNs. Reviewers question the unusually low AUC reported for the MIMIC-IV dataset and call for clarification.

Further issues include limited reproducibility due to the absence of shared code, and questions regarding the level of novelty and the choice of baselines for comparison. Despite these criticisms, the general sentiment is that the paper holds promise and could have a moderate-to-high impact if the authors address these substantial concerns. Overall, the work is seen as technically solid but in need of critical refinements and further justification to solidify its claimed contributions.